# Implicit Regularization in Deep Matrix Factorization

**Sanjeev Arora**
Princeton University and Institute for Advanced Study
arora@cs.princeton.edu

**Nadav Cohen**
Tel Aviv University
cohennadav@cs.tau.ac.il

**Wei Hu**
Princeton University
huwei@cs.princeton.edu

**Yuping Luo**
Princeton University
yupingl@cs.princeton.edu

## Abstract

Efforts to understand the generalization mystery in deep learning have led to the belief that gradient-based optimization induces a form of implicit regularization, a bias towards models of low "complexity." We study the implicit regularization of gradient descent over deep linear neural networks for matrix completion and sensing, a model referred to as deep matrix factorization. Our first finding, supported by theory and experiments, is that adding depth to a matrix factorization enhances an implicit tendency towards low-rank solutions, oftentimes leading to more accurate recovery. Secondly, we present theoretical and empirical arguments questioning a nascent view by which implicit regularization in matrix factorization can be captured using simple mathematical norms. Our results point to the possibility that the language of standard regularizers may not be rich enough to fully encompass the implicit regularization brought forth by gradient-based optimization.

## 1 Introduction

It is a mystery how deep neural networks generalize despite having far more learnable parameters than training examples. Explicit regularization techniques alone cannot account for this generalization, as they do not prevent the networks from being able to fit random data (see [52]). A view by which gradient-based optimization induces an *implicit regularization* has thus arisen. Of course, this view would be uninsightful if "implicit regularization" were treated as synonymous with "promoting generalization" — the question is whether we can characterize the implicit regularization independently of any validation data. Notably, the mere use of the term "regularization" already predisposes us towards characterizations based on known explicit regularizers (*e.g.* a constraint on some norm of the parameters), but one must also be open to the possibility that something else is afoot.

An old argument (*cf.* [25, 29]) traces implicit regularization in deep learning to beneficial effects of noise introduced by small-batch stochastic optimization. The feeling is that solutions that do not generalize correspond to "sharp minima," and added noise prevents convergence to such solutions. However, recent evidence (*e.g.* [26, 51]) suggests that deterministic (or near-deterministic) gradient-based algorithms can also generalize, and thus a different explanation is in order.

A major hurdle in this study is that implicit regularization in deep learning seems to kick in only with certain types of data (not with random data for example), and we lack mathematical tools for reasoning about real-life data. Thus one needs a simple test-bed for the investigation, where data admits a crisp mathematical formulation. Following earlier works, we focus on the problem of *matrix completion*: given a randomly chosen subset of entries from an unknown matrix $W^*$, the task is to recover the unseen entries. To cast this as a prediction problem, we may view each entry in $W^*$ as a data point: observed entries constitute the training set, and the average reconstruction error over the unobserved entries is the test error, quantifying generalization. Fitting the observed entries is obviously an underdetermined problem with multiple solutions. However, an extensive body of work (see [11] for a survey) has shown that if $W^*$ is low-rank, certain technical assumptions (*e.g.* "incoherence") are satisfied and sufficiently many entries are observed, then various algorithms

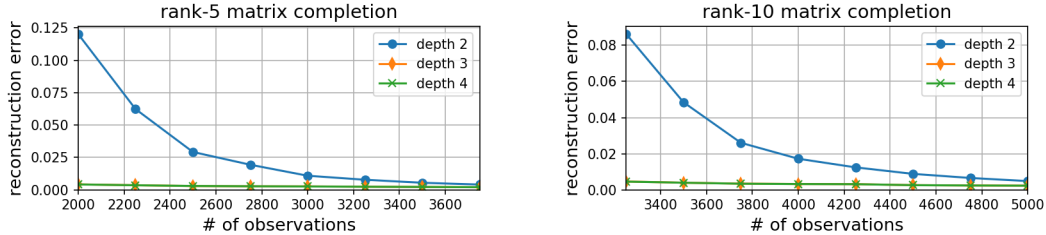

Figure 1: Matrix completion via gradient descent over deep matrix factorizations. Left (respectively, right) plot shows reconstruction errors for matrix factorizations of depths 2, 3 and 4, when applied to the completion of a random rank-5 (respectively, rank-10) matrix with size $100 \times 100$. $x$-axis stands for the number of observed entries (randomly chosen), $y$-axis represents reconstruction error, and error bars (indiscernible) mark standard deviations of the results over multiple trials. All matrix factorizations are full-dimensional, *i.e.* have hidden dimensions 100. Both learning rate and standard deviation of (random, zero-centered) initialization for gradient descent were set to the small value $10^{-3}$. Notice, with few observed entries factorizations of depths 3 and 4 significantly outperform that of depth 2, whereas with more entries all factorizations perform well. For further details, and a similar experiment on matrix sensing tasks, see Appendix E.

can achieve approximate or even exact recovery. Of these, a well-known method based upon convex optimization finds the minimal nuclear norm matrix among those fitting all observed entries (see [9]).[1]

One may try to solve matrix completion using shallow neural networks. A natural approach, *matrix factorization*, boils down to parameterizing the solution as a product of two matrices — $W = W_2 W_1$ — and optimizing the resulting (non-convex) objective for fitting observed entries. Formally, this can be viewed as training a depth-2 linear neural network. It is possible to explicitly constrain the rank of the produced solution by limiting the shared dimension of $W_1$ and $W_2$. However, in practice, even when the rank is unconstrained, running gradient descent with small learning rate (step size) and initialization close to zero tends to produce low-rank solutions, and thus allows accurate recovery if $W^*$ is low-rank. This empirical observation led Gunasekar *et al.* to conjecture in [20] that gradient descent over a matrix factorization induces an implicit regularization minimizing nuclear norm:

**Conjecture 1** (from [20], informally stated). *With small enough learning rate and initialization close enough to the origin, gradient descent on a full-dimensional matrix factorization converges to the minimum nuclear norm solution.*

**Deep matrix factorization**    Since standard matrix factorization can be viewed as a two-layer neural network (with linear activations), a natural extension is to consider deeper models. A *deep matrix factorization*[2] of $W \in \mathbb{R}^{d,d'}$, with hidden dimensions $d_1, \ldots, d_{N-1} \in \mathbb{N}$, is the parameterization:

$$W = W_N W_{N-1} \cdots W_1 \,, \tag{1}$$

where $W_j \in \mathbb{R}^{d_j, d_{j-1}}$, $j = 1, \ldots, N$, with $d_N := d$, $d_0 := d'$. $N$ is referred to as the *depth* of the factorization, the matrices $W_1, \ldots, W_N$ as its *factors*, and the resulting $W$ as the *product matrix*.

Could the implicit regularization of deep matrix factorizations be stronger than that of their shallow counterpart (which Conjecture 1 equates with nuclear norm minimization)? Experiments reported in Figure 1 suggest that this is indeed the case — depth leads to more accurate completion of a low-rank matrix when the number of observed entries is small. Our purpose in the current paper is to mathematically analyze this stronger form of implicit regularization. Can it be described by a matrix norm (or quasi-norm) continuing the line of Conjecture 1, or is a paradigm shift required?

## 1.1   Paper overview

Review of related work is given in Appendix A (deferred to supplementary material per lack of space).

In Section 2 we investigate the potential of norms for capturing the implicit regularization in deep matrix factorization. Surprisingly, we find that the main theoretical evidence connecting nuclear norm and shallow (depth-2) matrix factorization — proof given in [20] for Conjecture 1 in a particular restricted setting — extends to arbitrarily deep factorizations as well. This result disqualifies the

natural hypothesis by which Schatten quasi-norms replace nuclear norm as the implicit regularization when one adds depth to a shallow matrix factorization. Instead, when interpreted through the lens of [20], it brings forth a conjecture by which the implicit regularization is captured by nuclear norm for any depth. Since our experiments (Figure 1) show that depth changes (enhances) the implicit regularization, we are led to question the theoretical direction proposed in [20], and accordingly conduct additional experiments to evaluate the validity of Conjecture 1.

Typically, when the number of observed entries is sufficiently large with respect to the rank of the matrix to recover, nuclear norm minimization yields exact recovery, and thus it is impossible to distinguish between that and a different implicit regularization which also perfectly recovers. The regime most interesting to evaluate is therefore that in which the number of observed entries is too small for exact recovery by nuclear norm minimization — here there is room for different implicit regularizations to manifest themselves by providing higher quality solutions. Our empirical results show that in this regime, matrix factorizations consistently outperform nuclear norm minimization, suggesting that their implicit regularization admits stronger bias towards low-rank, in contrast to Conjecture 1. Together, our theory and experiments lead us to suspect that the implicit regularization in matrix factorization (shallow or deep) may not be amenable to description by a simple mathematical norm, and a detailed analysis of the dynamics in optimization may be necessary.

Section 3 carries out such an analysis, characterizing how the singular value decomposition of the learned solution evolves during gradient descent. Evolution rates of singular values turn out to be proportional to their size exponentiated by $2 - 2/N$, where $N$ is the depth of the factorization. This establishes a tendency towards low rank solutions, which intensifies with depth. Experiments validate the findings, demonstrating the dynamic nature of implicit regularization in deep matrix factorization.

We believe the trajectories traversed in optimization may be key to understanding generalization in deep learning, and hope that our work will inspire further progress along this line.

## 2 Can the implicit regularization be captured by norms?

In this section we investigate the possibility of extending Conjecture 1 for explaining implicit regularization in deep matrix factorization. Given the experimental evidence in Figure 1, one may hypothesize that gradient descent on a depth-$N$ matrix factorization implicitly minimizes some norm (or quasi-norm) that approximates rank, with the approximation being more accurate the larger $N$ is. For example, a natural candidate would be Schatten-$p$ quasi-norm to the power of $p$ ($0 < p \leq 1$), which for a matrix $W \in \mathbb{R}^{d,d'}$ is defined as: $\|W\|_{S_p}^p := \sum_{r=1}^{\min\{d,d'\}} \sigma_r^p(W)$, where $\sigma_1(W), \ldots, \sigma_{\min\{d,d'\}}(W)$ are the singular values of $W$. For $p = 1$ this reduces to nuclear norm, which by Conjecture 1 corresponds to a depth-2 factorization. As $p$ approaches zero we obtain a closer approximation of $\text{rank}(W)$, which could be suitable for factorizations of higher depths.

We will focus in this section on *matrix sensing* — a more general problem than matrix completion. Here, we are given $m$ measurement matrices $A_1, \ldots, A_m$, with corresponding labels $y_1, \ldots, y_m$ generated by $y_i = \langle A_i, W^* \rangle$, and our goal is to reconstruct the unknown matrix $W^*$. As in the case of matrix completion, well-known methods, and in particular nuclear norm minimization, can recover $W^*$ if it is low-rank, certain technical conditions are met, and sufficiently many observations are given (see [42]).

### 2.1 Current theory does not distinguish depth-$N$ from depth-2

Our first result is that the theory developed by [20] to support Conjecture 1 can be generalized to suggest that nuclear norm captures the implicit regularization in matrix factorization not just for depth 2, but for arbitrary depth. This is of course inconsistent with the experimental findings reported in Figure 1. We will first recall the existing theory, and then show how to extend it.

[20] studied implicit regularization in shallow (depth-2) matrix factorization by considering recovery of a positive semidefinite matrix from sensing via symmetric measurements, namely:

$$\min_{W \in \mathcal{S}_+^d} \ell(W) := \tfrac{1}{2} \sum_{i=1}^m (y_i - \langle A_i, W \rangle)^2, \qquad (2)$$

where $A_1, \ldots, A_m \in \mathbb{R}^{d,d}$ are symmetric and linearly independent, and $\mathcal{S}_+^d$ stands for the set of (symmetric and) positive semidefinite matrices in $\mathbb{R}^{d,d}$. Focusing on the underdetermined regime $m \ll d^2$, they investigated the implicit bias brought forth by running *gradient flow* (gradient descent with infinitesimally small learning rate) on a symmetric full-rank matrix factorization, *i.e.* on the objective:

$$\psi : \mathbb{R}^{d,d} \to \mathbb{R}_{\geq 0} \quad , \quad \psi(Z) := \ell(ZZ^\top) = \tfrac{1}{2}\sum\nolimits_{i=1}^{m}(y_i - \langle A_i, ZZ^\top\rangle)^2 \,.$$

For $\alpha > 0$, denote by $W_{\mathrm{sha},\infty}(\alpha)$ (sha here stands for "shallow") the final solution $ZZ^\top$ obtained from running gradient flow on $\psi(\cdot)$ with initialization $\alpha I$ ($\alpha$ times identity). Formally, $W_{\mathrm{sha},\infty}(\alpha) := \lim_{t\to\infty} Z(t)Z(t)^\top$ where $Z(0) = \alpha I$ and $\dot{Z}(t) = -\frac{d\psi}{dZ}(Z(t))$ for $t \in \mathbb{R}_{\geq 0}$ ($t$ here is a continuous time index, and $\dot{Z}(t)$ stands for the derivative of $Z(t)$ with respect to time). Letting $\|\cdot\|_*$ represent matrix nuclear norm, the following result was proven by [20]:

**Theorem 1** (adaptation of Theorem 1 in [20]). *Assume the measurement matrices $A_1, \ldots, A_m$ commute. Then, if $\bar{W}_{\mathrm{sha}} := \lim_{\alpha\to 0} W_{\mathrm{sha},\infty}(\alpha)$ exists and is a global optimum for Equation* (2) *with $\ell(\bar{W}_{\mathrm{sha}}) = 0$, it holds that $\bar{W}_{\mathrm{sha}} \in \mathrm{argmin}_{W\in\mathcal{S}_+^d, \, \ell(W)=0} \|W\|_*$, i.e. $\bar{W}_{\mathrm{sha}}$ is a global optimum with minimal nuclear norm.*[3]

Motivated by Theorem 1 and empirical evidence they provided, [20] raised Conjecture 1, which, formally stated, hypothesizes that the condition in Theorem 1 of $\{A_i\}_{i=1}^{m}$ commuting is unnecessary, and an identical statement holds for arbitrary (symmetric linearly independent) measurement matrices.[4]

While the analysis of [20] covers only symmetric matrix factorizations of the form $ZZ^\top$, they noted that it can be extended to also account for asymmetric factorizations of the type considered in the current paper. Specifically, running gradient flow on the objective:

$$\phi(W_1, W_2) := \ell(W_2 W_1) = \tfrac{1}{2}\sum\nolimits_{i=1}^{m}(y_i - \langle A_i, W_2 W_1\rangle)^2 \,,$$

with $W_1, W_2 \in \mathbb{R}^{d,d}$ initialized to $\alpha I$, $\alpha > 0$, and denoting by $W_{\mathrm{sha},\infty}(\alpha)$ the product matrix obtained at the end of optimization (*i.e.* $W_{\mathrm{sha},\infty}(\alpha) := \lim_{t\to\infty} W_2(t)W_1(t)$ where $W_j(0) = \alpha I$ and $\dot{W}_j(t) = -\frac{\partial\phi}{\partial W_j}(W_1(t), W_2(t))$ for $t \in \mathbb{R}_{\geq 0}$), Theorem 1 holds exactly as stated. For completeness, we provide a proof of this fact in Appendix D.

Next, we show that Theorem 1 — the main theoretical justification for the connection between nuclear norm and shallow matrix factorization — extends to arbitrarily deep factorizations as well. Consider gradient flow over the objective:

$$\phi(W_1, \ldots, W_N) := \ell(W_N W_{N-1} \cdots W_1) = \tfrac{1}{2}\sum\nolimits_{i=1}^{m}(y_i - \langle A_i, W_N W_{N-1} \cdots W_1\rangle)^2 \,,$$

with $W_1, \ldots, W_N \in \mathbb{R}^{d,d}$ initialized to $\alpha I$, $\alpha > 0$. Using $W_{\mathrm{deep},\infty}(\alpha)$ to denote the product matrix obtained at the end of optimization (*i.e.* $W_{\mathrm{deep},\infty}(\alpha) := \lim_{t\to\infty} W_N(t)W_{N-1}(t) \cdots W_1(t)$ where $W_j(0) = \alpha I$ and $\dot{W}_j(t) = -\frac{\partial\phi}{\partial W_j}(W_1(t), \ldots, W_N(t))$ for $t \in \mathbb{R}_{\geq 0}$), a result analogous to Theorem 1 holds:

**Theorem 2.** *Suppose $N \geq 3$, and that the matrices $A_1, \ldots, A_m$ commute. Then, if $\bar{W}_{\mathrm{deep}} := \lim_{\alpha\to 0} W_{\mathrm{deep},\infty}(\alpha)$ exists and is a global optimum for Equation* (2) *with $\ell(\bar{W}_{\mathrm{deep}}) = 0$, it holds that $\bar{W}_{\mathrm{deep}} \in \mathrm{argmin}_{W\in\mathcal{S}_+^d, \, \ell(W)=0} \|W\|_*$, i.e. $\bar{W}_{\mathrm{deep}}$ is a global optimum with minimal nuclear norm.*[5]

*Proof sketch (for complete proof see Appendix C.1).* Our proof is inspired by that of Theorem 1 given in [20]. Using the expression for $\dot{W}(t)$ derived in [3] (Lemma 3 in Appendix B), it can be shown that $W(t)$ commutes with $\{A_i\}_{i=1}^{m}$, and takes on a particular form. Taking limits $t \to \infty$ and $\alpha \to 0$, optimality (minimality) of nuclear norm is then established using a duality argument. $\quad\square$

Theorem 2 provides a particular setting where the implicit regularization in deep matrix factorizations boils down to nuclear norm minimization. By Proposition 1 below, there exist instances of this setting for which the minimization of nuclear norm contradicts minimization (even locally) of Schatten-$p$

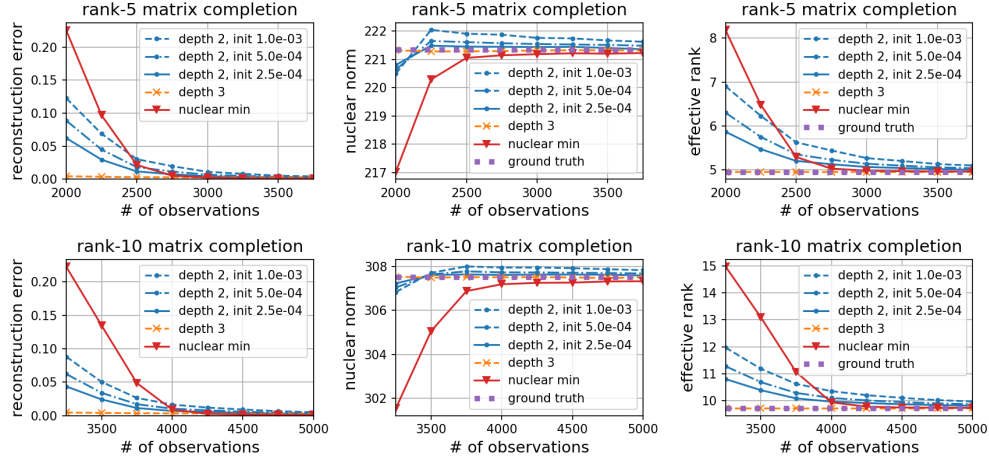

Figure 2: Evaluation of nuclear norm as the implicit regularization in deep matrix factorization. Each plot compares gradient descent over matrix factorizations of depths 2 and 3 (results for depth 4 were indistinguishable from those of depth 3; we omit them to reduce clutter) against minimum nuclear norm solution and ground truth in matrix completion tasks. Top (respectively, bottom) row corresponds to completion of a random rank-5 (respectively, rank-10) matrix with size $100 \times 100$. Left, middle and right columns display (in $y$-axis) reconstruction error, nuclear norm and effective rank (*cf.* [43]) respectively. In each plot, $x$-axis stands for the number of observed entries (randomly chosen), and error bars (indiscernible) mark standard deviations of the results over multiple trials. All matrix factorizations are full-dimensional, *i.e.* have hidden dimensions 100. Both learning rate and standard deviation of (random, zero-centered) initialization for gradient descent were initially set to $10^{-3}$. Running with smaller learning rate did not yield a noticeable change in terms of final results. Initializing with smaller standard deviation had no observable effect on results of depth 3 (and 4), but did impact those of depth 2 — the outcomes of dividing standard deviation by 2 and by 4 are included in the plots.[7] Notice, with many observed entries minimum nuclear norm solution coincides with ground truth (minimum rank solution), and matrix factorizations of all depths converge to these. On the other hand, when there are fewer observed entries minimum nuclear norm solution does not coincide with ground truth, and matrix factorizations prefer to lower the effective rank at the expense of higher nuclear norm, in a manner that is more potent for deeper factorizations. For further details, and a similar experiment on matrix sensing tasks, see Appendix E.

quasi-norm for any $0 < p < 1$. Therefore, one cannot hope to capture the implicit regularization in deep matrix factorizations through Schatten quasi-norms. Instead, if we interpret Theorem 2 through the lens of [20], we arrive at a conjecture by which the implicit regularization is captured by nuclear norm for any depth.

**Proposition 1.** *For any dimension $d \geq 3$, there exist linearly independent symmetric and commutable measurement matrices $A_1, \ldots, A_m \in \mathbb{R}^{d,d}$, and corresponding labels $y_1, \ldots, y_m \in \mathbb{R}$, such that the limit solution defined in Theorem 2 — $\bar{W}_{\mathrm{deep}}$ — which has been shown to satisfy $\bar{W}_{\mathrm{deep}} \in \mathrm{argmin}_{W \in \mathcal{S}_+^d, \ell(W)=0} \|W\|_*$, is not a local minimum of the following program for any $0 < p < 1$:[6]*

$$\min_{W \in \mathcal{S}_+^d, \ell(W)=0} \|W\|_{S_p} .$$

*Proof sketch (for complete proof see Appendix C.2).* We choose $A_1, \ldots, A_m$ and $y_1, \ldots, y_m$ such that: *(i)* $\bar{W}_{\mathrm{deep}} = \mathrm{diag}(1, 1, 0, \ldots, 0)$; and *(ii)* adding $\epsilon \in (0, 1)$ to entries $(1, 2)$ and $(2, 1)$ of $\bar{W}_{\mathrm{deep}}$ maintains optimality. The result then follows from the fact that the addition of $\epsilon$ decreases Schatten-$p$ quasi-norm for any $0 < p < 1$. □

## 2.2 Experiments challenging Conjecture 1

Subsection 2.1 suggests that from the perspective of current theory, it is natural to apply Conjecture 1 to matrix factorizations of arbitrary depth. On the other hand, the experiment reported in Figure 1 implies that depth changes (enhances) the implicit regularization. To resolve this tension we conduct a more refined experiment, which ultimately puts in question the validity of Conjecture 1.

Our experimental protocol is as follows. For different matrix completion tasks with varying number of observed entries, we compare minimum nuclear norm solution to those brought forth by running gradient descent on matrix factorizations of different depths. For each depth, we apply gradient descent with different choices of learning rate and standard deviation for (random, zero-centered) initialization, observing the trends as these become smaller. The outcome of the experiment is presented in Figure 2. As can be seen, when the number of observed entries is sufficiently large with respect to the rank of the matrix to recover, factorizations of all depths indeed admit solutions that tend to minimum nuclear norm. However, when there are less entries observed — precisely the data-poor setting where implicit regularization matters most — neither shallow (depth-2) nor deep (depth-$N$ with $N \geq 3$) factorizations minimize nuclear norm. Instead, they put more emphasis on lowering the effective rank (*cf.* [43]), in a manner which is stronger for deeper factorizations.

A close look at the experiments of [20] reveals that there too, in situations where the number of observed entries (or sensing measurements) was small (less than required for reliable recovery), a discernible gap appeared between the minimal nuclear norm and that returned by (gradient descent on) a matrix factorization. In light of Figure 2, we believe that if [20] had included in its plots an accurate surrogate for rank (*e.g.* effective rank or Schatten-$p$ quasi-norm with small $p$), scenarios where matrix factorization produced sub-optimal (higher than minimum) nuclear norm would have manifested superior (lower) rank. More broadly, our experiments suggest that the implicit regularization in (shallow or deep) matrix factorization is somehow geared towards low rank, and just so happens to minimize nuclear norm in cases with sufficiently many observations, where minimum nuclear norm and minimum rank are known to coincide (*cf.* [9, 42]). We note that the theoretical analysis of [32] supporting Conjecture 1 is limited to such cases, and thus cannot truly distinguish between nuclear norm minimization and some other form of implicit regularization favoring low rank.

Given that Conjecture 1 seems to hold in some settings (Theorems 1 and 2; [32]) but not in other (Figure 2), we hypothesize that capturing implicit regularization in (shallow or deep) matrix factorization through a single mathematical norm (or quasi-norm) may not be possible, and a detailed account for the optimization process might be necessary. This is carried out in Section 3.

## 3   Dynamical analysis

This section characterizes trajectories of gradient flow (gradient descent with infinitesimally small learning rate) on deep matrix factorizations. The characterization significantly extends past analyses for linear neural networks (*e.g.* [44, 3]) — we derive differential equations governing the dynamics of singular values and singular vectors for the product matrix $W$ (Equation (1)). Evolution rates of singular values turn out to be proportional to their size exponentiated by $2 - 2/N$, where $N$ is the depth of the factorization. For singular vectors, we show that lack of movement implies a particular form of alignment with the gradient, and by this strengthen past results which have only established the converse. Via theoretical and empirical demonstrations, we explain how our findings imply a tendency towards low-rank solutions, which intensifies with depth.

Our derivation treats a setting which includes matrix completion and sensing as special cases. We assume minimization of a general analytic loss $\ell(\cdot)$,[8] overparameterized by a deep matrix factorization:

$$\phi(W_1, \ldots, W_N) := \ell(W_N W_{N-1} \cdots W_1) . \tag{3}$$

We study gradient flow over the factorization:

$$\dot{W}_j(t) := \tfrac{d}{dt} W_j(t) = -\tfrac{\partial}{\partial W_j} \phi(W_1(t), \ldots, W_N(t)) \quad , \ t \geq 0 , \ j = 1, \ldots, N , \tag{4}$$

and in accordance with past work, assume that factors are *balanced* at initialization, *i.e.*:

$$W_{j+1}^\top(0) W_{j+1}(0) = W_j(0) W_j^\top(0) \quad , \ j = 1, \ldots, N-1 . \tag{5}$$

Equation (5) is satisfied approximately in the common setting of near-zero initialization (it holds exactly in the "residual" setting of identity initialization — *cf.* [23, 5]). The condition played an important role in the analysis of [3], facilitating derivation of a differential equation governing the product matrix of a linear neural network (see Lemma 3 in Appendix B). It was shown in [3] empirically that there is an excellent match between the theoretical predictions of gradient flow with balanced initialization, and its practical realization via gradient descent with small learning rate and near-zero initialization. Other works (*e.g.* [4, 28]) later supported this match theoretically.

We note that by Section 6 in [3], for depth $N \geq 3$, the dynamics of the product matrix $W$ (Equation (1)) *cannot* be exactly equivalent to gradient descent on the loss $\ell(\cdot)$ regularized by a penalty term. This preliminary observation already hints to the possibility that the effect of depth is different from those of standard regularization techniques.

Employing results of [3], we will characterize the evolution of singular values and singular vectors for $W$. As a first step, we show that $W$ admits an *analytic singular value decomposition* ([7, 12]):

**Lemma 1.** *The product matrix $W(t)$ can be expressed as:*

$$W(t) = U(t)S(t)V^\top(t), \tag{6}$$

*where: $U(t) \in \mathbb{R}^{d,\min\{d,d'\}}$, $S(t) \in \mathbb{R}^{\min\{d,d'\},\min\{d,d'\}}$ and $V(t) \in \mathbb{R}^{d',\min\{d,d'\}}$ are analytic functions of $t$; and for every $t$, the matrices $U(t)$ and $V(t)$ have orthonormal columns, while $S(t)$ is diagonal (elements on its diagonal may be negative and may appear in any order).*

*Proof sketch (for complete proof see Appendix C.3).* We show that $W(t)$ is an analytic function of $t$ and then invoke Theorem 1 in [7]. $\qquad\square$

The diagonal elements of $S(t)$, which we denote by $\sigma_1(t), \ldots, \sigma_{\min\{d,d'\}}(t)$, are signed singular values of $W(t)$; the columns of $U(t)$ and $V(t)$, denoted $\mathbf{u}_1(t), \ldots, \mathbf{u}_{\min\{d,d'\}}(t)$ and $\mathbf{v}_1(t), \ldots, \mathbf{v}_{\min\{d,d'\}}(t)$, are the corresponding left and right singular vectors (respectively).

With Lemma 1 in place, we are ready to characterize the evolution of singular values:

**Theorem 3.** *The signed singular values of the product matrix $W(t)$ evolve by:*

$$\dot{\sigma}_r(t) = -N \cdot \left(\sigma_r^2(t)\right)^{1-1/N} \cdot \left\langle \nabla\ell(W(t)), \mathbf{u}_r(t)\mathbf{v}_r^\top(t) \right\rangle \quad, \ r = 1, \ldots, \min\{d, d'\}. \tag{7}$$

*If the matrix factorization is non-degenerate, i.e. has depth $N \geq 2$, the singular values need not be signed (we may assume $\sigma_r(t) \geq 0$ for all t).*

*Proof sketch (for complete proof see Appendix C.4).* Differentiating the analytic singular value decomposition (Equation (6)) with respect to time, multiplying from the left by $U^\top(t)$ and from the right by $V(t)$, and using the fact that $U(t)$ and $V(t)$ have orthonormal columns, we obtain $\dot{\sigma}_r(t) = \mathbf{u}_r^\top(t)\dot{W}(t)\mathbf{v}_r(t)$. Equation (7) then follows from plugging in the expression for $\dot{W}(t)$ developed by [3] (Lemma 3 in Appendix B). $\qquad\square$

Strikingly, given a value for $W(t)$, the evolution of singular values depends on $N$ — depth of the matrix factorization — only through the multiplicative factors $N \cdot (\sigma_r^2(t))^{1-1/N}$ (see Equation (7)). In the degenerate case $N = 1$, *i.e.* when the product matrix $W(t)$ is simply driven by gradient flow over the loss $\ell(\cdot)$ (no matrix factorization), the multiplicative factors reduce to 1, and the singular values evolve by: $\dot{\sigma}_r(t) = -\left\langle \nabla\ell(W(t)), \mathbf{u}_r(t)\mathbf{v}_r^\top(t) \right\rangle$. With $N \geq 2$, *i.e.* when depth is added to the factorization, the multiplicative factors become non-trivial, and while the constant $N$ does not differentiate between singular values, the terms $(\sigma_r^2(t))^{1-1/N}$ do — they enhance movement of large singular values, and on the other hand attenuate that of small ones. Moreover, the enhancement/attenuation becomes more significant as $N$ (depth of the factorization) grows.

Next, we turn to the evolution of singular vectors:

**Lemma 2.** *Assume that at initialization, the singular values of the product matrix $W(t)$ are distinct and different from zero.[9] Then, its singular vectors evolve by:*

$$
\begin{aligned}
\dot{U}(t) \ = \ & -U(t)\left(F(t) \odot \left[U^\top(t)\nabla\ell(W(t))V(t)S(t) + S(t)V^\top(t)\nabla\ell^\top(W(t))U(t)\right]\right) \\
& - \left(I_d - U(t)U^\top(t)\right)\nabla\ell(W(t))V(t)(S^2(t))^{\frac{1}{2}-\frac{1}{N}}
\end{aligned} \tag{8}
$$

$$
\begin{aligned}
\dot{V}(t) \ = \ & -V(t)\left(F(t) \odot \left[S(t)U^\top(t)\nabla\ell(W(t))V(t) + V^\top(t)\nabla\ell^\top(W(t))U(t)S(t)\right]\right) \\
& - \left(I_{d'} - V(t)V^\top(t)\right)\nabla\ell^\top(W(t))U^\top(t)(S^2(t))^{\frac{1}{2}-\frac{1}{N}},
\end{aligned} \tag{9}
$$

*where $I_d$ and $I_{d'}$ are the identity matrices of sizes $d \times d$ and $d' \times d'$ respectively, $\odot$ stands for the Hadamard (element-wise) product, and the matrix $F(t) \in \mathbb{R}^{\min\{d,d'\},\min\{d,d'\}}$ is skew-symmetric with $((\sigma_{r'}^2(t))^{1/N} - (\sigma_r^2(t))^{1/N})^{-1}$ in its $(r, r')$'th entry, $r \neq r'$.[10]*

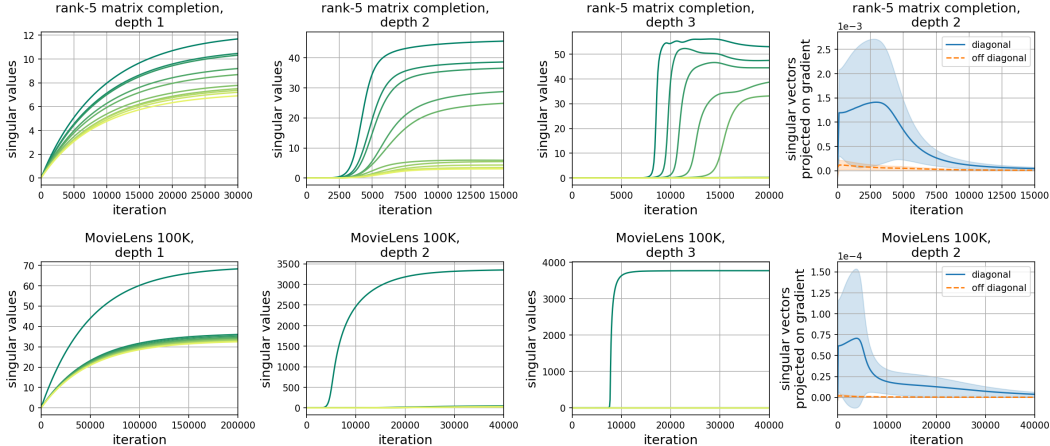

Figure 3: Dynamics of gradient descent over deep matrix factorizations — specifically, evolution of singular values and singular vectors of the product matrix during training for matrix completion. Top row corresponds to the task of completing a random rank-5 matrix with size $100 \times 100$ based on 2000 randomly chosen observed entries; bottom row corresponds to training on 10000 entries chosen randomly from the MovieLens 100K dataset (completion of a $943 \times 1682$ matrix, *cf.* [24]).[11] First (left) three columns show top singular values for, respectively, depths 1 (no matrix factorization), 2 (shallow matrix factorization) and 3 (deep matrix factorization). Last (right) column shows singular vectors for a depth-2 factorization, by comparing on- *vs.* off-diagonal entries in the matrix $U^\top(t)\nabla\ell(W(t))V(t)$ (see Corollary 1) — for each group of entries, mean of absolute values is plotted, along with shaded area marking the standard deviation. All matrix factorizations are full-dimensional (hidden dimensions 100 in top row plots, 943 in bottom row plots). Notice, increasing depth makes singular values move slower when small and faster when large (in accordance with Theorem 3), which results in solutions with effectively lower rank. Notice also that $U^\top(t)\nabla\ell(W(t))V(t)$ is diagonally dominant so long as there is movement, showing that singular vectors of the product matrix align with those of the gradient (in accordance with Corollary 1). For further details, and a similar experiment on matrix sensing, see Appendix E.

*Proof sketch (for complete proof see Appendix C.5).* We follow a series of steps adopted from [46] to obtain expressions for $\dot{U}(t)$ and $\dot{V}(t)$ in terms of $U(t)$, $V(t)$, $S(t)$ and $\dot{W}(t)$. Plugging in the expression for $\dot{W}(t)$ developed by [3] (Lemma 3 in Appendix B) then yields Equations (8), (9). ☐

**Corollary 1.** *Assume the conditions of Lemma 2, and that the matrix factorization is non-degenerate, i.e. has depth $N \geq 2$. Then, for any time $t$ such that the singular vectors of the product matrix $W(t)$ are stationary, i.e. $\dot{U}(t) = 0$ and $\dot{V}(t) = 0$, it holds that $U^\top(t)\nabla\ell(W(t))V(t)$ is diagonal, meaning they align with the singular vectors of $\nabla\ell(W(t))$.*

*Proof sketch (for complete proof see Appendix C.6).* By Equations (8) and (9), $U^\top(t)\dot{U}(t)S(t) - S(t)V^\top(t)\dot{V}(t)$ is equal to the Hadamard product between $U^\top(t)\nabla\ell(W(t))V(t)$ and a (time-dependent) square matrix with zeros on its diagonal and non-zeros elsewhere. When $\dot{U}(t) = 0$ and $\dot{V}(t) = 0$ obviously $U^\top(t)\dot{U}(t)S(t) - S(t)V^\top(t)\dot{V}(t) = 0$, and so the Hadamard product is zero. This implies that $U^\top(t)\nabla\ell(W(t))V(t)$ is diagonal. ☐

Earlier papers studying gradient flow for linear neural networks (*e.g.* [44, 1, 31]) could show that singular vectors are stationary if they align with the singular vectors of the gradient. Corollary 1 is significantly stronger and implies a converse — if singular vectors are stationary, they must be aligned with the gradient. Qualitatively, this suggests that a "goal" of gradient flow on a deep matrix factorization is to align singular vectors of the product matrix with those of the gradient.

## 3.1 Implicit regularization towards low rank

Figure 3 presents empirical demonstrations of our conclusions from Theorem 3 and Corollary 1. It shows that for a non-degenerate deep matrix factorization, *i.e.* one with depth $N \geq 2$, under gradient descent with small learning rate and near-zero initialization, singular values of the product matrix are subject to an enhancement/attenuation effect as described above: they progress very slowly after initialization, when close to zero; then, upon reaching a certain threshold, the movement of a singular

value becomes rapid, with the transition from slow to rapid movement being sharper with a deeper factorization (larger $N$). In terms of singular vectors, the figure shows that those of the product matrix indeed align with those of the gradient. Overall, the dynamics promote solutions that have a few large singular values and many small ones, with a gap that is more extreme the deeper the matrix factorization is. This is an implicit regularization towards low rank, which intensifies with depth.

**Theoretical illustration** Consider the simple case of square matrix sensing with a single measurement fit via $\ell_2$ loss: $\ell(W) = \frac{1}{2}(\langle A, W \rangle - y)^2$, where $A \in \mathbb{R}^{d,d}$ is the measurement matrix, and $y \in \mathbb{R}$ the corresponding label. Suppose we learn by running gradient flow over a depth-$N$ matrix factorization, *i.e.* over the objective $\phi(\cdot)$ defined in Equation (3). Corollary 1 states that the singular vectors of the product matrix — $\{\mathbf{u}_r(t)\}_r$ and $\{\mathbf{v}_r(t)\}_r$ — are stationary only when they diagonalize the gradient, meaning $\{|\mathbf{u}_r^\top(t)\nabla\ell(W(t))\mathbf{v}_r| : r = 1, \dots, d\}$ coincides with the set of singular values in $\nabla\ell(W(t))$. In our case $\nabla\ell(W) = (\langle A, W \rangle - y)A$, so stationarity of singular vectors implies $|\mathbf{u}_r^\top(t)\nabla\ell(W(t))\mathbf{v}_r| = |\delta(t)| \cdot \rho_r$, where $\delta(t) := \langle A, W(t) \rangle - y$ and $\rho_1, \dots, \rho_d$ are the singular values of $A$ (in no particular order). We will assume that starting from some time $t_0$ singular vectors are stationary, and accordingly $\mathbf{u}_r^\top(t)\nabla\ell(W(t))\mathbf{v}_r(t) = \delta(t) \cdot e_r \cdot \rho_r$ for $r = 1, \dots, d$, where $e_1, \dots, e_d \in \{-1, 1\}$. Theorem 3 then implies that (signed) singular values of the product matrix evolve by:

$$\dot{\sigma}_r(t) = -N \cdot \left(\sigma_r^2(t)\right)^{1-1/N} \cdot \delta(t) \cdot e_r \cdot \rho_r \quad , \forall t \geq t_0. \tag{10}$$

Let $r_1, r_2 \in \{1, \dots, d\}$. By Equation (10):

$$\int_{t'=t_0}^{t} \left(\sigma_{r_1}^2(t')\right)^{-1+1/N} \dot{\sigma}_{r_1}(t')dt' = \frac{e_{r_1}\rho_{r_1}}{e_{r_2}\rho_{r_2}} \cdot \int_{t'=t_0}^{t} \left(\sigma_{r_2}^2(t')\right)^{-1+1/N} \dot{\sigma}_{r_2}(t')dt'.$$

Computing the integrals, we may express $\sigma_{r_1}(t)$ as a function of $\sigma_{r_2}(t)$:[12]

$$\sigma_{r_1}(t) = \begin{cases} \alpha_{r_1,r_2} \cdot \sigma_{r_2}(t) + const & , N = 1 \\ \left(\sigma_{r_2}(t)\right)^{\alpha_{r_1,r_2}} \cdot const & , N = 2 \\ \left(\alpha_{r_1,r_2} \cdot \left(\sigma_{r_2}(t)\right)^{-\frac{N-2}{N}} + const\right)^{-\frac{N}{N-2}} & , N \geq 3 \end{cases}, \tag{11}$$

where $\alpha_{r_1,r_2} := e_{r_1}\rho_{r_1}(e_{r_2}\rho_{r_2})^{-1}$, and *const* stands for a value that does not depend on $t$. Equation 11 reveals a gap between $\sigma_{r_1}(t)$ and $\sigma_{r_2}(t)$ that enhances with depth. For example, consider the case where $0 < \alpha_{r_1,r_2} < 1$. If the depth $N$ is one, *i.e.* the matrix factorization is degenerate, $\sigma_{r_1}(t)$ will grow linearly with $\sigma_{r_2}(t)$. If $N = 2$ — shallow matrix factorization — $\sigma_{r_1}(t)$ will grow polynomially more slowly than $\sigma_{r_2}(t)$ (*const* here is positive). Increasing depth further will lead $\sigma_{r_1}(t)$ to asymptote when $\sigma_{r_2}(t)$ grows, at a value which can be shown to be lower the larger $N$ is. Overall, adding depth to the matrix factorization leads to more significant gaps between singular values of the product matrix, *i.e.* to a stronger implicit bias towards low rank.

## 4    Conclusion

The implicit regularization of gradient-based optimization is key to generalization in deep learning. As a stepping stone towards understanding this phenomenon, we studied deep linear neural networks for matrix completion and sensing, a model referred to as deep matrix factorization. Through theory and experiments, we questioned prevalent norm-based explanations for implicit regularization in matrix factorization (*cf.* [20]), and offered an alternative, dynamical approach. Our characterization of the dynamics induced by gradient flow on the singular value decomposition of the learned matrix significantly extends prior work on linear neural networks. It reveals an implicit tendency towards low rank which intensifies with depth, supporting the empirical superiority of deeper matrix factorizations.

An emerging view is that understanding optimization in deep learning necessitates a detailed account for the trajectories traversed in training (*cf.* [4]). Our work adds another dimension to the potential importance of trajectories — we believe they are necessary for understanding generalization as well, and in particular, may be key to analyzing implicit regularization for non-linear neural networks.

## Acknowledgments

This work was supported by NSF, ONR, Simons Foundation, Schmidt Foundation, Mozilla Research, Amazon Research, DARPA and SRC. Nadav Cohen was a member at the Institute for Advanced Study, and was additionally supported by the Zuckerman Israeli Postdoctoral Scholars Program. The authors thank Nathan Srebro for illuminating discussions which helped improve the paper.

## Footnotes

[1]Recall that the nuclear norm (also known as trace norm) of a matrix is the sum of its singular values, regarded as a convex relaxation of rank.

[2]Note that the literature includes various usages of this term — some in line with ours (*e.g.* [47, 53, 33]), while others less so (*e.g.* [50, 16, 49]).

[3] The result of [20] is slightly more general — it allows gradient flow to be initialized by $\alpha O$, where $O$ is an arbitrary orthogonal matrix, and it is shown that this leads to the exact same $W_{\mathrm{sha},\infty}(\alpha)$ as one would obtain from initializing at $\alpha I$. For simplicity, we limit our discussion to the latter initialization.

[4] Their conjecture also relaxes the requirement from the initialization of gradient flow — an initial value of $\alpha Z_0$ is believed to suffice, where $Z_0$ is an arbitrary full-rank matrix (that does not depend on $\alpha$).

[5] By Appendix C.1: $W_N(t)W_{N-1}(t)\cdots W_1(t) \succeq 0 \ \forall t$. Therefore, even though the theorem treats optimization over $\mathcal{S}_+^d$ using an unconstrained asymmetric factorization, gradient flow implicitly constrains the search to $\mathcal{S}_+^d$, so the assumption of $\bar{W}_{\mathrm{deep}}$ being a global optimum for Equation (2) with $\ell(\bar{W}_{\mathrm{deep}}) = 0$ is no stronger than the analogous assumption in Theorem 1 from [20]. The implicit constraining to $\mathcal{S}_+^d$ also holds when $N = 2$ (see Appendix D), so the asymmetric extension of Theorem 1 does not involve strengthening assumptions either.

[6]Following [20], we take for granted existence of $\bar{W}_{\mathrm{deep}}$ and it being a global optimum for Equation (2) with $\ell(\bar{W}_{\mathrm{deep}}) = 0$. If this is not the case then Theorem 2 does not apply, and hence it obviously does not disqualify minimization of Schatten quasi-norms as the implicit regularization.

[7]As can be seen, using smaller initialization enhanced the implicit tendency of depth-2 matrix factorization towards low rank. It is possible that this tendency can eventually match that of depth-3 (and -4), but only if initialization size goes far below what is customary in deep learning.

[8]A function $f(\cdot)$ is *analytic* on a domain $\mathcal{D}$ if at every $x \in \mathcal{D}$: it is infinitely differentiable; and its Taylor series converges to it on some neighborhood of $x$ (see [30] for further details).

[9]This assumption can be relaxed significantly — all that is needed is that no singular value be identically zero ($\forall r \ \exists t \ s.t. \ \sigma_r(t) \neq 0$), and no pair of singular values be identical through time ($\forall r, r' \ \exists t \ s.t. \ \sigma_r(t) \neq \sigma_{r'}(t)$).

[10]Equations (8) and (9) are well-defined when $t$ is such that $\sigma_1(t), \ldots, \sigma_{\min\{d,d'\}}(t)$ are distinct and different from zero. By analyticity, this is either the case for every $t$ besides a set of isolated points, or it is not the case for any $t$. Our assumption on initialization disqualifies the latter option, so any $t$ for which Equations (8) or (9) are ill-defined is isolated. The derivatives of $U$ and $V$ for such $t$ may thus be inferred by continuity.

[11]Observations of MovieLens 100K were subsampled solely for reducing run-time.

[12]In accordance with Theorem 3, if $N \geq 2$, we assume without loss of generality that $\sigma_{r_1}(t), \sigma_{r_2}(t) \geq 0$, while disregarding the trivial case of equality to zero.

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
