[Supplementary Material · reg_dmf_supp.pdf]

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

[13][1] and [19] also considered settings where there are multiple global minima, but in these too there was just one solution to which optimization could converge, leaving only the question of what path is taken to reach it.

[14]In addition to standard linear neural networks, [22] also analyzed "linear convolutional networks", characterized by a particular weight sharing pattern. For such models, the implicit regularization was found to promote sparsity in the frequency domain, in a manner which does depend on depth.

[15] Existence of $\lim_{t \to \infty} \tilde{W}(t)$ implies that $\lim_{t \to \infty} \tilde{\mathcal{A}}^\dagger (\mathbf{s}(t))$ exists (see Equation (16)), which in turn, given the linear independence of $\{\tilde{A}_i\}_{i=1}^m$, indicates that $\lim_{t \to \infty} \mathbf{s}(t)$ exists as well. Similarly to [20], the remainder of the proof treats the case where $\lim_{t \to \infty} \mathbf{s}(t)$ is finite, thereby avoiding the technical load associated with infinite coordinates.

[16]Note that for $W \in \mathcal{S}^d_+$ we have $\|W\|_* = \langle I, W \rangle$.

[17]In the proof of Theorem 2 (Appendix C.1), diagonality of $A_1, \ldots, A_m$ corresponds to the case where $O$ — the diagonalizing matrix — is simply the identity, and therefore $\bar{W}_{\mathrm{deep}}$ is equal to $\tilde{W}_{\mathrm{deep}}^*$, implying that the former is indeed diagonal and positive semidefinite.

[18]To see this, note that $U^\top(t)U(t)$ is constant, thus its derivative with respect to time is equal to zero, i.e. $\dot{U}^\top(t)U(t) + U^\top(t)\dot{U}(t) = 0$ (by an analogous argument $\dot{V}^\top(t)V(t) + V^\top(t)\dot{V}(t) = 0$ holds as well).

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

# A   Related work

Implicit regularization in deep learning is a highly active area of research. For non-linear neural networks, the topic has thus far been studied empirically (*e.g.* in [37, 52, 29, 26, 38]), with theoretical analyses being somewhat scarce (see [15, 41] for some of the few observations that have been derived). The majority of theoretical attention has been devoted to (single-layer) linear predictors and (multi-layer) linear neural networks, often viewed as stepping stones towards non-linear models. Linear predictors were treated in [34, 45, 36, 21]. For linear neural networks, [1, 31, 19] studied settings where the training objective admits a single global minimum, and the question is what path gradient descent (or gradient flow) takes to reach it.[13] This stands in contrast to the practical deep learning scenario where there are multiple global minima, and implicit regularization refers to the optimizer being biased towards reaching those solutions that generalize well. The latter scenario was treated by [22] and [28] in the context of linear neural networks trained for binary classification via separable data. These works showed that under certain assumptions, gradient descent converges (in direction) to the maximum margin solution. Intriguingly, the bias towards maximum margin holds with any number of layers, so in particular, implicit regularization was found to be oblivious to depth.[14]

The most extensively studied instance of linear neural networks is matrix factorization, corresponding to a model with multiple inputs, multiple outputs and a single hidden layer, typically trained to recover a low-rank linear mapping. The literature on matrix factorization for low-rank matrix recovery is far too broad to cover here — we refer to [10] for a recent survey, while mentioning that the technique is oftentimes attributed to [8]. Notable works proving successful recovery of a low-rank matrix through matrix factorization trained by gradient descent with no explicit regularization are [48, 35, 32]. Of these, [32] can be viewed as resolving the conjecture of [20] — which we investigate in Section 2 — for the case of sufficiently many linear measurements satisfying the restricted isometry property.

To the best of our knowledge, the current paper is the first to study implicit regularization for deep (three or more layer) linear neural networks with multiple outputs. The latter trait seems to be distinctive, as it is the main differentiator between the setting of [22, 28], where implicit regularization is oblivious to depth, and ours, for which we show that depth has significant impact. We note that our work is focused on the type of solutions reached by gradient descent, not the complementary questions of whether an optimal solution is found, and how fast that happens. These questions were studied extensively for matrix factorization — *cf.* [17, 6, 39, 18] — and more recently for linear neural networks of arbitrary depth — see [5, 3, 4, 14]. From a technical perspective, closest to our work are [20] and [3] — we rely on their results and significantly extend them (see Sections 2 and 3).

# B   Useful lemmas

We recall the following result from [3], which characterizes the evolution of the product matrix under gradient flow on a deep matrix factorization:

**Lemma 3** (adaptation of Theorem 1 in [3]). *Let $\ell : \mathbb{R}^{d,d'} \to \mathbb{R}_{\geq 0}$ be a continuously differentiable loss, overparameterized by a deep matrix factorization:*

$$\phi(W_1, \ldots, W_N) = \ell(W_N W_{N-1} \cdots W_1).$$

*Suppose we run gradient flow over the factorization:*

$$\dot{W}_j(t) := \frac{d}{dt} W_j(t) = -\frac{\partial}{\partial W_j} \phi(W_1(t), \ldots, W_N(t)) \quad , t \geq 0 , j = 1, \ldots, N,$$

*with factors initialized to be balanced, i.e.:*

$$W_{j+1}^\top(0) W_{j+1}(0) = W_j(0) W_j^\top(0) \quad , j = 1, \ldots, N-1.$$

*Then, the product matrix $W(t) = W_N(t) \cdots W_1(t)$ obeys the following dynamics:*

$$\dot{W}(t) = -\sum_{j=1}^{N} \left[ W(t) W^\top(t) \right]^{\frac{j-1}{N}} \cdot \nabla \ell(W(t)) \cdot \left[ W^\top(t) W(t) \right]^{\frac{N-j}{N}},$$

*where $[\,\cdot\,]^\alpha$, $\alpha \in \mathbb{R}_{\geq 0}$, stands for a power operator defined over positive semidefinite matrices (with $\alpha = 0$ yielding identity by definition).*

An additional result we will use is the following technical lemma:

**Lemma 4.** *Let $\alpha \geq \frac{1}{2}$ and $g : [0, \infty) \to \mathbb{R}$ be a continuous function. Consider the initial value problem:*

$$s(0) = s_0 \quad , \quad \dot{s}(t) = (s^2(t))^\alpha \cdot g(t) \quad \forall t \geq 0, \tag{12}$$

*where $s_0 \in \mathbb{R}$. Then, as long as it does not diverge to $\pm\infty$, the solution to this problem ($s(t)$) has the same sign as its initial value ($s_0$). That is, $s(t)$ is identically zero if $s_0 = 0$, is positive if $s_0 > 0$, and is negative if $s_0 < 0$.*

*Proof.* If $\alpha = 1/2$, the solution to Equation (12) is:

$$s(t) = \begin{cases} s_0 \cdot \exp\left( \int_{t'=0}^{t} g(t')dt' \right) & , s_0 > 0 \\ s_0 \cdot \exp\left( -\int_{t'=0}^{t} g(t')dt' \right) & , s_0 < 0 \\ 0 & , s_0 = 0 \end{cases}.$$

This solution does not diverge in finite time (regardless of the chosen $g(\cdot)$), and obviously preserves the sign of its initial value.

If $\alpha > 1/2$, Equation (12) is solved by:

$$s(t) = \begin{cases} \left( s_0^{-2\alpha+1} + (-2\alpha+1) \int_{t'=0}^{t} g(t')dt' \right)^{\frac{1}{-2\alpha+1}} & , s_0 > 0 \\ -\left( (-s_0)^{-2\alpha+1} - (-2\alpha+1) \int_{t'=0}^{t} g(t')dt' \right)^{\frac{1}{-2\alpha+1}} & , s_0 < 0 \\ 0 & , s_0 = 0 \end{cases}.$$

In this case, divergence in finite time can take place (depending on the choice of $g(\cdot)$), but nonetheless the sign of $s(t)$ is preserved until that happens. $\qquad\square$

## C   Deferred proofs

### C.1   Proof of Theorem 2

For convenience, throughout the proof we replace the notation $\bar{W}_{\text{deep}}$ by $W_{\text{deep}}^*$. We also define a linear operator $\mathcal{A}$ which specifies all $m$ measurements:

$$\mathcal{A} : \mathbb{R}^{d,d} \to \mathbb{R}^m \quad , \quad \mathcal{A}(W) = \begin{pmatrix} \langle A_1, W \rangle \\ \vdots \\ \langle A_m, W \rangle \end{pmatrix},$$

and its adjoint operator $\mathcal{A}^\dagger$:

$$\mathcal{A}^\dagger : \mathbb{R}^m \to \mathbb{R}^{d,d} \quad , \quad \mathcal{A}^\dagger(\mathbf{r}) = \sum_{i=1}^{m} r_i A_i .$$

Then we can rewrite the loss function in Equation (2) as:

$$\ell(W) = \frac{1}{2} \left\| \mathcal{A}(W) - \mathbf{y} \right\|_2^2 ,$$

where $\mathbf{y} := (y_1, \dots, y_m)^\top \in \mathbb{R}^m$. The gradient of $\ell(\cdot)$ can be expressed as:

$$\nabla \ell(W) = \mathcal{A}^\dagger(\mathcal{A}(W) - \mathbf{y}).$$

We consider a fixed $\alpha > 0$ for now, and will take the limit $\alpha \to 0^+$ later. Recall that gradient flow is run on the objective $\phi(W_1, \dots, W_N) = \ell(W_N \cdots W_1)$, with initialization $W_j(0) = \alpha I$, $j =$

$1, \dots, N$. From Lemma 3, we know that the product matrix $W(t) = W_N(t) \cdots W_1(t)$ evolves by:

$$\dot{W}(t) = -\sum_{j=1}^{N} \left[ W(t)W^\top(t) \right]^{\frac{j-1}{N}} \cdot \nabla \ell \left( W(t) \right) \cdot \left[ W^\top(t)W(t) \right]^{\frac{N-j}{N}}$$

$$= -\sum_{j=1}^{N} \left[ W(t)W^\top(t) \right]^{\frac{j-1}{N}} \cdot \mathcal{A}^\dagger (\mathbf{r}(t)) \cdot \left[ W^\top(t)W(t) \right]^{\frac{N-j}{N}} \quad , \; t \in \mathbb{R}_{\geq 0} \,, \qquad (13)$$

$$W(0) = \alpha^N I \,,$$

where $\mathbf{r}(t) := \mathcal{A}(W(t)) - \mathbf{y}$ is the vector of residuals at time $t$. Since $A_1, \dots, A_m$ are symmetric and commutable, they are simultaneously (orthogonally) diagonalizable, *i.e.* there exists an orthogonal matrix $O \in \mathbb{R}^{d,d}$ such that $\tilde{A}_i := O A_i O^\top$, $i = 1, \dots, m$, are all diagonal. Consider a change of variables $\tilde{W}(t) := O W(t) O^\top$, and denote $\tilde{\mathcal{A}}^\dagger(\mathbf{r}) := O \mathcal{A}^\dagger(\mathbf{r}) O^\top = \sum_{i=1}^{m} r_i \tilde{A}_i$. Then it follows from Equation (13) that:

$$\dot{\tilde{W}}(t) = -\sum_{j=1}^{N} \left[ \tilde{W}(t)\tilde{W}^\top(t) \right]^{\frac{j-1}{N}} \cdot \tilde{\mathcal{A}}^\dagger (\mathbf{r}(t)) \cdot \left[ \tilde{W}^\top(t)\tilde{W}(t) \right]^{\frac{N-j}{N}} \quad , \; t \in \mathbb{R}_{\geq 0} \,,$$

$$\tilde{W}(0) = \alpha^N I \,. \qquad (14)$$

Notice that: *(i)* $\tilde{W}(0)$ is diagonal; and *(ii)* if $\tilde{W}(t)$ is diagonal then so is $\dot{\tilde{W}}(t)$. We may therefore set the off-diagonal elements of $\tilde{W}(t)$ to zero, and solve for the diagonal ones:

$$\dot{\tilde{W}}_{kk}(t) = -N \cdot \left( \tilde{W}_{kk}^2(t) \right)^{\frac{N-1}{N}} \cdot \tilde{\mathcal{A}}_{kk}^\dagger (\mathbf{r}(t)) \,, \quad \tilde{W}_{kk}(0) = \alpha^N \,, \; t \in \mathbb{R}_{\geq 0} \,, \; k = 1, \dots, d \,. \quad (15)$$

By Lemma 4, $\tilde{W}_{kk}(t)$ maintains the sign of its initialization, meaning it stays positive. Moreover, since by assumption $N \geq 3$, the solution to Equation (15) is:

$$\tilde{W}_{kk}(t) = \alpha^N \left( 1 + (N-2)\alpha^{N-2} \cdot \tilde{\mathcal{A}}_{kk}^\dagger (\mathbf{s}(t)) \right)^{-\frac{N}{N-2}} \,, \quad t \in \mathbb{R}_{\geq 0} \,, \quad k = 1, \dots, d \,,$$

where $\mathbf{s}(t) := \int_{t'=0}^{t} (\mathbf{r}(t')) dt'$. The matrix $\tilde{W}(t)$ thus has positive elements on its diagonal (and zeros elsewhere), and takes the following form:

$$\tilde{W}(t) = \alpha^N \left[ I_d + (N-2)\alpha^{N-2} \cdot \tilde{\mathcal{A}}^\dagger (\mathbf{s}(t)) \right]^{-\frac{N}{N-2}} \,, \qquad (16)$$

where $I_d$ is the $d \times d$ identity matrix, and $[\,\cdot\,]^{-N/(N-2)}$ is a negative power operator defined over positive definite matrices. We assume $W_{\text{deep},\infty}(\alpha) := \lim_{t \to \infty} W(t)$ exists, and so we may write:

$$\tilde{W}_{\text{deep},\infty}(\alpha) := O \, W_{\text{deep},\infty}(\alpha) \, O^\top = \lim_{t \to \infty} \tilde{W}(t) = \alpha^N \left[ I_d - \tilde{\mathcal{A}}^\dagger (\boldsymbol{\nu}_\infty(\alpha)) \right]^{-\frac{N}{N-2}} \,,$$

where $\boldsymbol{\nu}_\infty(\alpha) := -(N-2)\alpha^{N-2} \lim_{t \to \infty} \mathbf{s}(t)$.[15] Since $\{\tilde{W}(t)\}_t$ are diagonal, $\tilde{W}_{\text{deep},\infty}(\alpha)$ is diagonal. Additionally, positive definiteness of $\{\tilde{W}(t)\}_t$ implies that $\tilde{W}_{\text{deep},\infty}(\alpha)$ is positive definite as well (it cannot have zero eigenvalues as it is given by a negative power operator). This means that:

$$I_d - \tilde{\mathcal{A}}^\dagger (\boldsymbol{\nu}_\infty(\alpha)) = \alpha^{-N} \left[ \tilde{W}_{\text{deep},\infty}(\alpha) \right]^{-\frac{N-2}{N}} \succ 0 \implies \tilde{\mathcal{A}}^\dagger (\boldsymbol{\nu}_\infty(\alpha)) \prec I_d \,. \qquad (17)$$

Now take the limit $\alpha \to 0^+$. By assumption $W_{\text{deep}}^* := \lim_{\alpha \to 0^+} W_{\text{deep},\infty}(\alpha)$ exists, so we can write:

$$\tilde{W}_{\text{deep}}^* := O \, W_{\text{deep}}^* \, O^\top = \lim_{\alpha \to 0^+} \tilde{W}_{\text{deep},\infty}(\alpha) = \lim_{\alpha \to 0^+} \alpha^N \left[ I_d - \tilde{\mathcal{A}}^\dagger (\boldsymbol{\nu}_\infty(\alpha)) \right]^{-\frac{N}{N-2}} \,,$$

The fact that $\{\tilde{W}_{\text{deep},\infty}(\alpha)\}_\alpha$ are diagonal and positive definite implies that:

$$\tilde{W}_{\text{deep}}^* \succeq 0 \,. \qquad (18)$$

Moreover, if the $k$'th element on the diagonal of $\tilde{W}^*_{\text{deep}}$ is non-zero, it must hold that:

$$\lim_{\alpha \to 0^+} \alpha^N \left( 1 - \tilde{\mathcal{A}}^{\dagger}_{kk}\left( \boldsymbol{\nu}_{\infty}(\alpha) \right) \right)^{-\frac{N}{N-2}} \neq 0 \implies \lim_{\alpha \to 0^+} \tilde{\mathcal{A}}^{\dagger}_{kk}\left( \boldsymbol{\nu}_{\infty}(\alpha) \right) = 1\,,$$

from which we conclude:

$$\left\langle I_d - \tilde{\mathcal{A}}^{\dagger}\left( \boldsymbol{\nu}_{\infty}(\alpha) \right), \tilde{W}^*_{\text{deep}} \right\rangle = \sum_{k=1}^{d} \left( 1 - \tilde{\mathcal{A}}^{\dagger}_{kk}\left( \boldsymbol{\nu}_{\infty}(\alpha) \right) \right) \cdot \left( \tilde{W}^*_{\text{deep}} \right)_{kk} \xrightarrow[\alpha \to 0^+]{} 0\,. \qquad (19)$$

Returning to the original variables (un-diagonalizing by the orthogonal matrix $O$), recall that:

$$W^*_{\text{deep}} = O^{\top}\, \tilde{W}^*_{\text{deep}}\, O\,, \qquad (20)$$

and:

$$\mathcal{A}^{\dagger}\left( \boldsymbol{\nu}_{\infty}(\alpha) \right) = O^{\top} \tilde{\mathcal{A}}^{\dagger}\left( \boldsymbol{\nu}_{\infty}(\alpha) \right) O\,, \qquad (21)$$

therefore the following hold:

- $W^*_{\text{deep}} \succeq 0$  (by Equations (18) and (20));
- $\mathcal{A}(W^*_{\text{deep}}) = \mathbf{y}$  (by assumption);
- $\forall \alpha > 0 : \mathcal{A}^{\dagger}\left( \boldsymbol{\nu}_{\infty}(\alpha) \right) \prec I_d$  (by Equations (17) and (21)); and
- $\lim_{\alpha \to 0^+} \left\langle I_d - \tilde{\mathcal{A}}^{\dagger}\left( \boldsymbol{\nu}_{\infty}(\alpha) \right), \tilde{W}^*_{\text{deep}} \right\rangle = 0$  (by Equations (19), (20) and (21)).

Lemma 5 below then concludes the proof. $\qquad \square$

**Lemma 5.** *Suppose that $W^* \in \mathcal{S}^d_+$ satisfies $\mathcal{A}(W^*) = \mathbf{y}$, and that there exists a sequence of vectors $\boldsymbol{\nu}_1, \boldsymbol{\nu}_2, \ldots \in \mathbb{R}^m$ such that $\mathcal{A}^{\dagger}(\boldsymbol{\nu}_n) \preceq I$ for all $n$ and $\lim_{n \to \infty} \langle I - \mathcal{A}^{\dagger}(\boldsymbol{\nu}_n), W^* \rangle = 0$. Then $W^* \in \operatorname{argmin}_{W \in \mathcal{S}^d_+,\, \mathcal{A}(W) = \mathbf{y}} \|W\|_*$.*

*Proof.* Recall that $\mathcal{S}^d_+$ stands for the set of (symmetric and) positive semidefinite matrices in $\mathbb{R}^{d,d}$, and $\|\cdot\|_*$ denotes matrix nuclear norm. The minimization problem being considered can be framed as a semidefinite program:[16]

$$\begin{aligned} \text{minimize} \quad & \langle I, W \rangle \\ \text{subject to} \quad & \mathcal{A}(W) = \mathbf{y} \\ & W \in \mathcal{S}^d_+\,. \end{aligned} \qquad (22)$$

A corresponding dual program is:

$$\begin{aligned} \text{maximize} \quad & \boldsymbol{\nu}^{\top}\mathbf{y} \\ \text{subject to} \quad & \mathcal{A}^*(\boldsymbol{\nu}) \preceq I \\ & \boldsymbol{\nu} \in \mathbb{R}^m\,. \end{aligned} \qquad (23)$$

Let OPT be the optimal value for the primal program (Equation (22)):

$$\text{OPT} := \min_{W \in \mathcal{S}^d_+,\, \mathcal{A}(W) = \mathbf{y}} \|W\|_*\,.$$

By duality theory, for any $\boldsymbol{\nu}$ feasible in the dual program (Equation (23)), we have $\boldsymbol{\nu}^{\top}\mathbf{y} \leq \text{OPT}$. Since $W^*$ is feasible in the primal, and each $\boldsymbol{\nu}_n$ is feasible in the dual, it holds that:

$$0 \leq \|W^*\|_* - \text{OPT} \leq \|W^*\|_* - \boldsymbol{\nu}_n^{\top}\mathbf{y} = \langle I, W^* \rangle - \boldsymbol{\nu}_n^{\top}\mathcal{A}(W^*) = \langle I - \mathcal{A}^{\dagger}(\boldsymbol{\nu}_n), W^* \rangle\,.$$

Taking the limit $n \to \infty$, the right hand side above becomes $0$, which implies $\|W^*\|_* = \text{OPT}$. $\quad \square$

## C.2 Proof of Proposition 1

We will choose $A_1, \ldots, A_m$ to be diagonal. This of course ensures symmetry and commutativity. Additionally, by the proof of Theorem 2 (Appendix C.1), it implies that $\bar{W}_{\mathrm{deep}}$ is diagonal and positive semidefinite.[17] We set $A_1, \ldots, A_m$ and $y_1, \ldots, y_m$ such that the linear equations $\langle A_i, W \rangle = y_i$, $i = 1, \ldots, m$, are the following:

$$
\begin{aligned}
W_{11} &= W_{22} \\
W_{11} &= W_{kk} + 1 \quad , \ k = 3, 4, \ldots, d.
\end{aligned}
\tag{24}
$$

Note that the matrices $A_1, \ldots, A_m$ which naturally induce these equations are (diagonal and) linearly independent, as required. We know that $\bar{W}_{\mathrm{deep}}$ is diagonal and has minimal nuclear norm among all positive semidefinite matrices that satisfy the equations. Using the fact that for positive semidefinite matrices nuclear norm is the same as trace, one readily sees that:

$$
\bar{W}_{\mathrm{deep}} = \mathrm{diag}(1, 1, 0, 0, \ldots, 0).
$$

We complete the proof by showing that in any neighborhood of $\bar{W}_{\mathrm{deep}}$, there exists a positive semidefinite matrix that meets Equation (24) and has strictly smaller Schatten-$p$ quasi-norm for any $0 < p < 1$. Indeed, for $\epsilon \in (0, 1)$ define:

$$
\hat{W}_\epsilon := \begin{pmatrix}
1 & \epsilon & 0 & \cdots & 0 \\
\epsilon & 1 & 0 & \cdots & 0 \\
0 & 0 & 0 & \cdots & 0 \\
\vdots & \vdots & \vdots & \ddots & \vdots \\
0 & 0 & 0 & \cdots & 0
\end{pmatrix} \in \mathbb{R}^{d,d}.
$$

$\hat{W}_\epsilon$ obviously satisfies Equation (24). Additionally, it is symmetric with eigenvalues:

$$
\lambda_1 = 1 + \epsilon \ , \ \lambda_2 = 1 - \epsilon \ , \ \lambda_3 = \cdots = \lambda_d = 0,
$$

and therefore is positive semidefinite. For any $0 < p < 1$:

$$
\|\hat{W}_\epsilon\|_{S_p}^p = (1 - \epsilon)^p + (1 + \epsilon)^p < 2 \cdot \left( \tfrac{1}{2}(1 + \epsilon) + \tfrac{1}{2}(1 - \epsilon) \right)^p = 2 = \|\bar{W}_{\mathrm{deep}}\|_{S_p}^p,
$$

where the inequality follows from $\theta_p : \mathbb{R}_{\geq 0} \to \mathbb{R}_{\geq 0}$, $\theta_p(x) = x^p$, being strictly concave. Noting that taking $\epsilon \to 0^+$ makes $\hat{W}_\epsilon$ arbitrarily close to $\bar{W}_{\mathrm{deep}}$, we conclude the proof. $\qquad \square$

## C.3 Proof of Lemma 1

By Theorem 1 in [7], it suffices to show that the product matrix $W(t)$ is an analytic function of $t$. Analytic functions are closed under summation, multiplication and composition, so the analyticity of $\ell(\cdot)$ implies that $\phi(\cdot)$ (Equation (3)) is analytic as well. It then follows (see Theorem 1.1 in [27]) that under gradient flow (Equation (4)), the factors $W_1(t), \ldots, W_N(t)$ are analytic functions of $t$. Therefore $W(t)$ (Equation (1)) is also analytic in $t$. $\qquad \square$

## C.4 Proof of Theorem 3

Differentiate the analytic singular value decomposition (Equation (6)) with respect to time:

$$
\dot{W}(t) = \dot{U}(t) S(t) V^\top(t) + U(t) \dot{S}(t) V^\top(t) + U(t) S(t) \dot{V}^\top(t),
$$

then multiply from the left by $U^\top(t)$ and from the right by $V(t)$:

$$
U^\top(t) \dot{W}(t) V(t) = U^\top(t) \dot{U}(t) S(t) + \dot{S}(t) + S(t) \dot{V}^\top(t) V(t),
$$

where we used the fact that $U(t)$ and $V(t)$ have orthonormal columns. Restricting our attention to the diagonal elements of this matrix equation, we have:

$$
\mathbf{u}_r^\top(t) \dot{W}(t) \mathbf{v}_r(t) = \langle \mathbf{u}_r(t), \dot{\mathbf{u}}_r(t) \rangle \sigma_r(t) + \dot{\sigma}_r(t) + \sigma_r(t) \langle \dot{\mathbf{v}}_r(t), \mathbf{v}_r(t) \rangle \quad , \ r = 1, \ldots, \min\{d, d'\}.
$$

Since $\mathbf{u}_r(t)$ has constant (unit) length it holds that $\langle \mathbf{u}_r(t), \dot{\mathbf{u}}_r(t) \rangle = \frac{1}{2} \cdot \frac{d}{dt} \|\mathbf{u}_r(t)\|_2^2 = 0$, and similarly $\langle \dot{\mathbf{v}}_r(t), \mathbf{v}_r(t) \rangle = 0$. The latter equation thus simplifies to:

$$\dot{\sigma}_r(t) = \mathbf{u}_r^\top(t) \dot{W}(t) \mathbf{v}_r(t) \quad , \; r = 1, \ldots, \min\{d, d'\}. \tag{25}$$

Lemma 3 from Appendix B provides the following expression for $\dot{W}(t)$:

$$\dot{W}(t) = -\sum_{j=1}^N \left[ W(t)W^\top(t) \right]^{\frac{j-1}{N}} \cdot \nabla\ell\big(W(t)\big) \cdot \left[ W^\top(t)W(t) \right]^{\frac{N-j}{N}},$$

where $[\,\cdot\,]^\alpha$, $\alpha \in \mathbb{R}_{\geq 0}$, stands for a power operator defined over positive semidefinite matrices (with $\alpha = 0$ yielding identity by definition). Plugging in the analytic singular value decomposition (Equation (6)) gives:

$$
\begin{aligned}
\dot{W}(t) \;=\; &-\nabla\ell\big(W(t)\big) \cdot V(t)\big(S^2(t)\big)^{\frac{N-1}{N}} V^\top(t) \\
&-\sum_{j=2}^{N-1} U(t)\big(S^2(t)\big)^{\frac{j-1}{N}} U^\top(t) \cdot \nabla\ell\big(W(t)\big) \cdot V(t)\big(S^2(t)\big)^{\frac{N-j}{N}} V^\top(t) \\
&-U(t)\big(S^2(t)\big)^{\frac{N-1}{N}} U^\top(t) \cdot \nabla\ell\big(W(t)\big).
\end{aligned}
$$

Left-multiplying by $\mathbf{u}_r^\top(t)$, right-multiplying by $\mathbf{v}_r(t)$, and using the fact that $\{\mathbf{u}_r(t)\}_r$ (columns of $U(t)$) and $\{\mathbf{v}_r(t)\}_r$ (columns of $V(t)$) are orthonormal sets, we obtain:

$$
\begin{aligned}
\mathbf{u}_r^\top(t)\dot{W}(t)\mathbf{v}_r(t) \;=\; &-\mathbf{u}_r^\top(t)\nabla\ell\big(W(t)\big)\mathbf{v}_r(t) \cdot (\sigma_r^2(t))^{\frac{N-1}{N}} \\
&-\sum_{j=2}^{N-1} (\sigma_r^2(t))^{\frac{j-1}{N}} \cdot \mathbf{u}_r^\top(t)\nabla\ell\big(W(t)\big)\mathbf{v}_r(t) \cdot (\sigma_r^2(t))^{\frac{N-j}{N}} \\
&-(\sigma_r^2(t))^{\frac{N-1}{N}} \cdot \mathbf{u}_r^\top(t)\nabla\ell\big(W(t)\big)\mathbf{v}_r(t) \\
\;=\; &-N \cdot (\sigma_r^2(t))^{\frac{N-1}{N}} \cdot \mathbf{u}_r^\top(t)\nabla\ell\big(W(t)\big)\mathbf{v}_r(t).
\end{aligned}
$$

Combining this with Equation (25) yields the sought-after Equation (7).

To complete the proof, it remains to show that if the matrix factorization is non-degenerate (has depth $N \geq 2$), singular values need not be signed, *i.e.* we may assume $\sigma_r(t) \geq 0$ for all $t$. Equation (7), along with Lemma 4, imply that if $N \geq 2$, $\sigma_r(t)$ will never switch sign. Therefore, either $\sigma_r(t) \geq 0$ for all $t$, or alternatively, this will hold if we take away a minus sign from $\sigma_r(t)$ and absorb it into $\mathbf{u}_r(t)$ (or $\mathbf{v}_r(t)$). $\qquad\square$

### C.5 Proof of Lemma 2

A real analytic function is either identically zero, or admits a zero set with no accumulation points (*cf.* [30]). For any $r \in \{1, \ldots, \min\{d, d'\}\}$, applying this fact to the signed singular value $\sigma_r(t)$, while taking into account our assumption of it being different from zero at initialization, we conclude that the set of times $t$ for which it vanishes has no accumulation points. Similarly, for any $r, r' \in \{1, \ldots, \min\{d, d'\}\}$, $r \neq r'$, we assumed that $\sigma_r^2(t) - \sigma_{r'}^2(t)$ is different from zero at initialization, and thus the set of times $t$ for which it vanishes is free from accumulation points. Overall, any time $t$ for which $\sigma_r(t) = 0$ for some $r$, or $\sigma_r^2(t) = \sigma_{r'}^2(t)$ for some $r \neq r'$, must be isolated, *i.e.* surrounded by a neighborhood in which none of these conditions are met. Accordingly, hereafter, we assume $\forall r : \sigma_r(t) \neq 0$ and $\forall r \neq r' : \sigma_r^2(t) \neq \sigma_{r'}^2(t)$, knowing that for times $t$ in which this does not hold, $\dot{U}(t)$ and $\dot{V}(t)$ can be inferred by continuity.

We now follow a series of steps adopted from [46], to derive expressions for $\dot{U}(t)$ and $\dot{V}(t)$ in terms of $U(t), V(t), S(t)$ and $\dot{W}(t)$. Differentiate the analytic singular value decomposition (Equation (6)) with respect to time:

$$\dot{W}(t) = \dot{U}(t)S(t)V^\top(t) + U(t)\dot{S}(t)V^\top(t) + U(t)S(t)\dot{V}^\top(t). \tag{26}$$

Multiplying from the left by $U^\top(t)$ and from the right by $V(t)$, we have:

$$U^\top(t)\dot{W}(t)V(t) = U^\top(t)\dot{U}(t)S(t) + \dot{S}(t) + S(t)\dot{V}^\top(t)V(t), \tag{27}$$

where we used the fact that $U(t)$ and $V(t)$ have orthonormal columns. This orthonormality also implies that $U^\top(t)\dot{U}(t)$ and $\dot{V}^\top(t)V(t)$ are skew-symmetric,[18] and in particular have zero diagonals. Since $S(t)$ is diagonal, $U^\top(t)\dot{U}(t)S(t)$ and $S(t)\dot{V}^\top(t)V(t)$ have zero diagonals as well. On the other hand $\dot{S}(t)$ holds zeros outside its diagonal, and so we may write:

$$\bar{I}_{\min\{d,d'\}} \odot (U^\top(t)\dot{W}(t)V(t)) = U^\top(t)\dot{U}(t)S(t) + S(t)\dot{V}^\top(t)V(t)\,, \tag{28}$$

where $\odot$ stands for Hadamard (element-wise) product, and $\bar{I}_{\min\{d,d'\}}$ is a $\min\{d,d'\} \times \min\{d,d'\}$ matrix holding zeros on its diagonal and ones elsewhere. Taking transpose of Equation (28), while recalling that $U^\top(t)\dot{U}(t)$ and $\dot{V}^\top(t)V(t)$ are skew-symmetric, we have:

$$\bar{I}_{\min\{d,d'\}} \odot (V^\top(t)\dot{W}^\top(t)U(t)) = -S(t)U^\top(t)\dot{U}(t) - \dot{V}^\top(t)V(t)S(t)\,. \tag{29}$$

Right-multiply Equation (28) by $S(t)$, left-multiply Equation (29) by $S(t)$, and add:

$$\bar{I}_{\min\{d,d'\}} \odot (U^\top(t)\dot{W}(t)V(t)S(t) + S(t)V^\top(t)\dot{W}^\top(t)U(t)) = U^\top(t)\dot{U}(t)S^2(t) - S^2(t)U^\top(t)\dot{U}(t)\,.$$

Since we assume diagonal elements of $S^2(t)$ are distinct ($\sigma_r^2(t) \neq \sigma_{r'}^2(t)$ for $r \neq r'$), this implies:

$$U^\top(t)\dot{U}(t) = H(t) \odot \left[ U^\top(t)\dot{W}(t)V(t)S(t) + S(t)V^\top(t)\dot{W}^\top(t)U(t) \right]\,,$$

where the matrix $H(t) \in \mathbb{R}^{\min\{d,d'\},\min\{d,d'\}}$ is defined by:

$$H_{r,r'}(t) := \begin{cases} \left(\sigma_{r'}^2(t) - \sigma_r^2(t)\right)^{-1} & ,r \neq r' \\ 0 & ,r = r' \end{cases}\,. \tag{30}$$

Multiplying from the left by $U(t)$ yields:

$$P_{U(t)}\dot{U}(t) = U(t)\left(H(t) \odot \left[U^\top(t)\dot{W}(t)V(t)S(t) + S(t)V^\top(t)\dot{W}^\top(t)U(t)\right]\right)\,, \tag{31}$$

with $P_{U(t)} := U(t)U^\top(t)$ being the projection onto the subspace spanned by the (orthonormal) columns of $U(t)$. Denote by $P_{U_\perp(t)}$ the projection onto the orthogonal complement, i.e. $P_{U_\perp(t)} := I_d - U(t)U^\top(t)$, where $I_d$ is the $d \times d$ identity matrix. Apply $P_{U_\perp(t)}$ to both sides of Equation (26):

$$P_{U_\perp(t)}\dot{W}(t) = P_{U_\perp(t)}\dot{U}(t)S(t)V^\top(t) + P_{U_\perp(t)}U(t)\dot{S}(t)V^\top(t) + P_{U_\perp(t)}U(t)S(t)\dot{V}^\top(t)\,.$$

Note that $P_{U_\perp(t)}U(t) = 0$, and multiply from the right by $V(t)S^{-1}(t)$ (the latter is well-defined since we assume diagonal elements of $S(t)$ are non-zero — $\sigma_r(t) \neq 0$):

$$P_{U_\perp(t)}\dot{U}(t) = P_{U_\perp(t)}\dot{W}(t)V(t)S^{-1}(t) = \left(I_d - U(t)U^\top(t)\right)\dot{W}(t)V(t)S^{-1}(t)\,. \tag{32}$$

Adding Equations (31) and (32), we obtain an expression for $\dot{U}(t)$:

$$\begin{aligned} \dot{U}(t) &= P_{U(t)}\dot{U}(t) + P_{U_\perp(t)}\dot{U}(t) \\ &= U(t)\left(H(t) \odot \left[U^\top(t)\dot{W}(t)V(t)S(t) + S(t)V^\top(t)\dot{W}^\top(t)U(t)\right]\right) \\ &\quad + \left(I_d - U(t)U^\top(t)\right)\dot{W}(t)V(t)S^{-1}(t)\,. \end{aligned} \tag{33}$$

By returning to Equations (28) and (29), switching the directions from which they were multiplied by $S(t)$ (i.e. multiplying Equation (28) from the left and Equation (29) from the right), and continuing similarly to above, an analogous expression for $\dot{V}(t)$ is derived:

$$\begin{aligned} \dot{V}(t) &= V(t)\left(H(t) \odot \left[S(t)U^\top(t)\dot{W}(t)V(t) + V^\top(t)\dot{W}^\top(t)U(t)S(t)\right]\right) \\ &\quad + \left(I_{d'} - V(t)V^\top(t)\right)\dot{W}^\top(t)U(t)S^{-1}(t)\,, \end{aligned} \tag{34}$$

where $I_{d'}$ is the $d' \times d'$ identity matrix.

Next, we invoke Lemma 3 from Appendix B, which provides an expression for $\dot{W}(t)$:

$$\dot{W}(t) = -\sum_{j=1}^{N} \left[W(t)W^\top(t)\right]^{\frac{j-1}{N}} \cdot \nabla\ell\big(W(t)\big) \cdot \left[W^\top(t)W(t)\right]^{\frac{N-j}{N}}\,, \tag{35}$$

where $[\,\cdot\,]^\alpha$, $\alpha \in \mathbb{R}_{\geq 0}$, stands for a power operator defined over positive semidefinite matrices (with $\alpha = 0$ yielding identity by definition). Plug the analytic singular value decomposition (Equation (6)) into Equation (35):

$$
\begin{aligned}
\dot{W}(t) \;=\; & -\nabla\ell\big(W(t)\big) \cdot V(t)\big(S^2(t)\big)^{\frac{N-1}{N}} V^\top(t) \\
& - \sum_{j=2}^{N-1} U(t)\big(S^2(t)\big)^{\frac{j-1}{N}} U^\top(t) \cdot \nabla\ell\big(W(t)\big) \cdot V(t)\big(S^2(t)\big)^{\frac{N-j}{N}} V^\top(t) \\
& - U(t)\big(S^2(t)\big)^{\frac{N-1}{N}} U^\top(t) \cdot \nabla\ell\big(W(t)\big)\,.
\end{aligned}
\tag{36}
$$

From this it follows that:

$$
\begin{aligned}
U^\top(t)\dot{W}(t)V(t) \;=\; & -U^\top(t)\nabla\ell\big(W(t)\big)V(t)\big(S^2(t)\big)^{\frac{N-1}{N}} \\
& - \sum_{j=2}^{N-1} \big(S^2(t)\big)^{\frac{j-1}{N}} U^\top(t)\nabla\ell\big(W(t)\big)V(t)\big(S^2(t)\big)^{\frac{N-j}{N}} \\
& - \big(S^2(t)\big)^{\frac{N-1}{N}} U^\top(t)\nabla\ell\big(W(t)\big)V(t) \\
\;=\; & -G(t) \odot \big[U^\top(t)\nabla\ell\big(W(t)\big)V(t)\big]\,,
\end{aligned}
\tag{37}
$$

where $G(t) \in \mathbb{R}^{\min\{d,d'\},\min\{d,d'\}}$ is defined by:

$$
G_{r,r'}(t) := \sum_{j=1}^{N} (\sigma_r^2(t))^{\frac{j-1}{N}} (\sigma_{r'}^2(t))^{\frac{N-j}{N}}\,.
\tag{38}
$$

Since $G(t)$ is symmetric (and $S(t)$ is diagonal), Equation (37) implies:

$$
\begin{aligned}
& U^\top(t)\dot{W}(t)V(t)S(t) + S(t)V^\top(t)\dot{W}^\top(t)U(t) \\
& = -G(t) \odot \big[U^\top(t)\nabla\ell\big(W(t)\big)V(t)S(t) + S(t)V^\top(t)\nabla\ell^\top\big(W(t)\big)U(t)\big]\,.
\end{aligned}
$$

Taking Hadamard product by $H(t)$ (Equation (30)) we obtain:

$$
\begin{aligned}
& H(t) \odot \big[U^\top(t)\dot{W}(t)V(t)S(t) + S(t)V^\top(t)\dot{W}^\top(t)U(t)\big] \\
& = -F(t) \odot \big[U^\top(t)\nabla\ell\big(W(t)\big)V(t)S(t) + S(t)V^\top(t)\nabla\ell^\top\big(W(t)\big)U(t)\big]\,,
\end{aligned}
$$

where $F(t) := H(t) \odot G(t)$ is given by:

$$
F_{r,r'}(t) := \begin{cases} \big((\sigma_{r'}^2(t))^{1/N} - (\sigma_r^2(t))^{1/N}\big)^{-1} & ,\, r \neq r' \\ 0 & ,\, r = r' \end{cases}\,.
\tag{39}
$$

Plug this into Equation (33):

$$
\begin{aligned}
\dot{U}(t) \;=\; & -U(t)\big(F(t) \odot \big[U^\top(t)\nabla\ell\big(W(t)\big)V(t)S(t) + S(t)V^\top(t)\nabla\ell^\top\big(W(t)\big)U(t)\big]\big) \\
& + \big(I_d - U(t)U^\top(t)\big)\dot{W}(t)V(t)S^{-1}(t)\,.
\end{aligned}
\tag{40}
$$

The first term on the right-hand side here complies with the result we seek to prove (Equation (8)). To treat the second term, we again invoke Equation (36), noting that the matrix $P_{U_\perp(t)} := I_d - U(t)U^\top(t)$ (projection onto the orthogonal complement of the subspace spanned by the columns of $U(t)$) produces zero when right-multiplied by $U(t)$. This implies:

$$
\big(I_d - U(t)U^\top(t)\big)\dot{W}(t)V(t)S^{-1}(t) = -\big(I_d - U(t)U^\top(t)\big)\nabla\ell\big(W(t)\big)V(t)\big(S^2(t)\big)^{\frac{1}{2}-\frac{1}{N}}\,.
$$

Plugging this back into Equation (40) yields Equation (8) — sought-after result. The analogous Equation (9) can be derived in a similar fashion (by incorporating Equation (35) into Equation (34), as we have done for Equation (33)). $\qquad\square$

## C.6 Proof of Corollary 1

As stated in the proof of Lemma 2 (Appendix C.5), for all $t$ but a set of isolated points it holds that $\forall r : \sigma_r(t) \neq 0$ and $\forall r \neq r' : \sigma_r^2(t) \neq \sigma_{r'}^2(t)$, meaning Equations (8) and (9) are well-defined. We

will initially assume this to be the case, and then treat isolated points by taking limits. Left-multiply Equation (8) by $U^\top(t)$ and Equation (9) by $V^\top(t)$:

$$U^\top(t)\dot{U}(t) = -F(t) \odot \left[ U^\top(t)\nabla\ell(W(t))V(t)S(t) + S(t)V^\top(t)\nabla\ell^\top(W(t))U(t) \right]$$
$$V^\top(t)\dot{V}(t) = -F(t) \odot \left[ S(t)U^\top(t)\nabla\ell(W(t))V(t) + V^\top(t)\nabla\ell^\top(W(t))U(t)S(t) \right],$$

where we have used the fact that $U(t)$ and $V(t)$ have orthonormal columns. Right-multiplying the first equation by $S(t)$, left-multiplying the second by $S(t)$, and then subtracting, we obtain:

$$U^\top(t)\dot{U}(t)S(t) - S(t)V^\top(t)\dot{V}(t)$$
$$= -F(t) \odot \left[ U^\top(t)\nabla\ell(W(t))V(t)S^2(t) - S^2(t)U^\top(t)\nabla\ell(W(t))V(t) \right]$$
$$= -F(t) \odot E(t) \odot \left[ U^\top(t)\nabla\ell(W(t))V(t) \right],$$

where the matrix $E(t) \in \mathbb{R}^{\min\{d,d'\},\min\{d,d'\}}$ is defined by: $E_{r,r'}(t) := \sigma_{r'}^2(t) - \sigma_r^2(t)$. Recalling the definition of $F(t)$ (Equation (39)), we have:

$$U^\top(t)\dot{U}(t)S(t) - S(t)V^\top(t)\dot{V}(t) = -\bar{I}_{\min\{d,d'\}} \odot G(t) \odot \left[ U^\top(t)\nabla\ell(W(t))V(t) \right], \quad (41)$$

where $G(t) \in \mathbb{R}^{\min\{d,d'\},\min\{d,d'\}}$ is the matrix defined in Equation (38), and $\bar{I}_{\min\{d,d'\}}$ is a matrix of the same size, with zeros on its diagonal and ones elsewhere. Since by assumption $\forall r : \sigma_r(t) \neq 0$, the matrix $G(t)$ does not contain zero elements. Therefore when $\dot{U}(t) = 0$ and $\dot{V}(t) = 0$, leading the left-hand side of Equation (41) to vanish, it must be that $U^\top(t)\nabla\ell(W(t))V(t)$ is diagonal.

To complete the proof, it remains to treat those isolated times $t$ for which the conditions $\forall r : \sigma_r(t) \neq 0$ and $\forall r \neq r' : \sigma_r^2(t) \neq \sigma_{r'}^2(t)$ do not all hold, and thus our derivation of Equation (41) may be invalid. Since both sides of the equation are continuous, it carries over to such isolated times, and is in fact applicable to any $t$. Accordingly, any $t$ for which $\dot{U}(t) = 0$ and $\dot{V}(t) = 0$ admits $\bar{I}_{\min\{d,d'\}} \odot G(t) \odot \left[ U^\top(t)\nabla\ell(W(t))V(t) \right] = 0$. Recalling the definition of $G(t)$ (Equation (38)), it is clear that the latter equality implies diagonality of $U^\top(t)\nabla\ell(W(t))V(t)$ if $\forall r : \sigma_r(t) \neq 0$. This means that the sought-after result holds if $\forall r : \sigma_r(t) \neq 0$ for every $t$. Recollecting that the product matrix is initialized to be full-rank ($\forall r : \sigma_r(0) \neq 0$), and invoking our assumption on the factorization being non-degenerate ($N \geq 2$), we apply Lemma 4 (from Appendix C.4) to the evolution of $\{\sigma_r(t)\}_r$ (Equation (7) in Theorem 3) and conclude the proof. $\qquad\square$

## D  Extension of [20] to asymmetric matrix factorization

Extending Theorem 1 from [20] to asymmetric (depth-2) matrix factorizations boils down to proving the following proposition:

**Proposition 2.** *Consider gradient flow on the objective:*

$$\phi(W_1, W_2) = \ell(W_2 W_1) = \frac{1}{2} \sum_{i=1}^m (y_i - \langle A_i, W_2 W_1 \rangle)^2,$$

*with $W_1, W_2 \in \mathbb{R}^{d,d}$ initialized to $\alpha I$, $\alpha > 0$, and denote by $W_{\mathrm{sha},\infty}(\alpha)$ the product matrix obtained at the end of optimization (i.e. $W_{\mathrm{sha},\infty}(\alpha) := \lim_{t\to\infty} W_2(t)W_1(t)$ where $W_j(0) = \alpha I$ and $\dot{W}_j(t) = -\frac{\partial\phi}{\partial W_j}(W_1(t), W_2(t))$ for $t \in \mathbb{R}_{\geq 0}$). Assume the measurement matrices $A_1, \ldots, A_m$ commute. Then, if $\bar{W}_{\mathrm{sha}} := \lim_{\alpha\to 0} W_{\mathrm{sha},\infty}(\alpha)$ exists and is a global optimum for Equation (2) with $\ell(\bar{W}_{\mathrm{sha}}) = 0$, it holds that $\bar{W}_{\mathrm{sha}} \in \mathrm{argmin}_{W\in\mathcal{S}_+^d, \ell(W)=0} \|W\|_*$, i.e. $\bar{W}_{\mathrm{sha}}$ is a global optimum with minimal nuclear norm.*

*Proof.* We follow the proof of Theorem 2 (Appendix C.1) up until Equation (15). Equation (13), specialized to $N = 2$, yields dynamics for the product matrix $W(t) = W_2(t)W_1(t)$:

$$\dot{W}(t) = -\mathcal{A}^*(\mathbf{r}(t)) \cdot \left[ W^\top(t)W(t) \right]^{\frac{1}{2}} - \left[ W(t)W^\top(t) \right]^{\frac{1}{2}} \cdot \mathcal{A}^*(\mathbf{r}(t)),$$
$$W(0) = \alpha^2 I.$$
$$\quad (42)$$

Figure 4: Matrix sensing via gradient descent over deep matrix factorizations. This figure is identical to Figure 1, except that reconstruction of a ground truth matrix is based not on a randomly chosen subset of entries, but on a set of random projections (*i.e.* on $\{\langle A_i, W^* \rangle\}_{i=1}^m$ where $W^*$ is the ground truth and $A_1, \ldots, A_m$ are measurement matrices drawn independently from a Gaussian distribution). For further details on this experiment see Appendix E.2.

Figure 5: Evaluation of nuclear norm as the implicit regularization in deep matrix factorization on matrix sensing tasks. This figure is identical to Figure 2, except that reconstruction of a ground truth matrix is based not on a randomly chosen subset of entries, but on a set of random projections (*i.e.* on $\{\langle A_i, W^* \rangle\}_{i=1}^m$ where $W^*$ is the ground truth and $A_1, \ldots, A_m$ are measurement matrices drawn independently from a Gaussian distribution). For further details on this experiment see Appendix E.2.

Equation (15), along with Lemma 4, imply that $\tilde{W}_{kk}(t)$ maintains the sign of its initialization, *i.e.* is positive. The diagonal matrix $\tilde{W}(t)$ is therefore positive definite, and so is the product matrix $W(t) = O^\top \tilde{W}(t) O$. Equation (42) thus becomes:

$$\dot{W}(t) = -\mathcal{A}^*(\mathbf{r}(t)) \cdot W(t) - W(t) \cdot \mathcal{A}^*(\mathbf{r}(t)),$$
$$W(0) = \alpha^2 I.$$
(43)

The dynamics in Equation (43) are precisely those developed in [20] for a symmetric matrix factorization. The proof of Theorem 1 there can now be applied as is, establishing the desired result. $\square$

# E    Further experiments and implementation details

## E.1    Further experiments

Figures 4, 5 and 6 present matrix sensing experiments supplementing the matrix completion experiments reported in Figures 1, 2 and 3 respectively.

Figure 6: Dynamics of gradient descent over deep matrix factorizations on a matrix sensing task. This figure is identical to the top row of Figure 3, except that training is based not on 2000 randomly chosen entries of the ground truth matrix, but on 2000 random projections (*i.e.* on $\{\langle A_i, W^* \rangle\}_{i=1}^{2000}$ where $W^*$ is the ground truth and $A_1, \ldots, A_{2000}$ are measurement matrices drawn independently from a Gaussian distribution). For further details on this experiment see Appendix E.2.

## E.2 Implementation details

In this appendix we provide implementation details omitted from the descriptions of our experiments (Figures 1, 2, 3, 4, 5 and 6). Our implementation is based on Python, with PyTorch ([40]) for realizing deep matrix factorizations and CVXPY ([13, 2]) for finding minimum nuclear norm solutions. Source code for reproducing our results can be found in `https://github.com/roosephu/deep_matrix_factorization`.

When referring to a random rank-$r$ matrix with size $d \times d'$, we mean a product $UV^\top$, where the entries of $U \in \mathbb{R}^{d,r}$ and $V \in \mathbb{R}^{d',r}$ are drawn independently from the standard normal distribution. Randomly chosen observed entries in synthetic matrix completion tasks (Figures 1, 2 and top row of Figure 3) were selected uniformly (without repetition). In synthetic matrix sensing tasks (Figures 4, 5 and 6), entries of all measurement (projection) matrices were drawn independently from the standard normal distribution. When varying the number of observations in synthetic matrix completion and sensing (Figures 1, 2, 4 and 5), we evaluated increments of 250. Training on MovieLens 100K dataset (bottom row of Figure 3) comprised fitting 10000 randomly (uniformly) chosen samples from the 100000 entries given in the $943 \times 1682$ user-movie rating matrix (see [24]).

In all experiments, deep matrix factorizations were trained by (full batch) gradient descent applied to $\ell_2$ loss over the observed entries (in matrix completion tasks) or given projections (in matrix sensing tasks), with no explicit regularization. Gradient descent was initialized by independently sampling all weights from a Gaussian distribution with zero mean and configurable standard deviation. Learning rates were fixed throughout optimization, and the stopping criterion was training loss reaching value lower than $10^{-6}$ (or $10^6$ iterations elapsing). In the nuclear norm evaluation experiments (Figures 2 and 5), learning rate and standard deviation of initialization for gradient descent were assigned values from the set $\{10^{-3}, 5 \cdot 10^{-4}, 2.5 \cdot 10^{-4}\}$. In the dynamics illustration experiments (Figures 3 and 6), displayed results correspond to both learning rate and standard deviation for initialization being $10^{-3}$.

In figures 1, 2, 4 and 5, each error bar marks standard deviation of the respective result over three trials differing in random seed for initialization of gradient descent. Reconstruction error with respect to a ground truth matrix $W^*$ is based on normalized Frobenius distance, *i.e.* for a solution $W$ it is $\|W - W^*\|_F / \|W^*\|_F$. In experiments with matrix completion and sensing under varying number of observations (Figures 1, 2, 4 and 5), plots begin at the smallest number for which stable results were obtained, and end when all evaluated methods are close to zero reconstruction error. For the dynamics illustration experiments (Figures 3 and 6), plots showing singular values hold 10 curves corresponding to the largest ones.