[Reviews · NeurIPS 2019]

Reviewer 1



I think this paper is a very good read. It is both pedagogical and brings interesting food for thought on a very active topic. English usage is good and references are adequate. Although it may be interesting to hint at how much the ideas could or could not be conveyed to more general nonlinear settings, the methodology is interesting and I particularly liked the core section 3 about dynamical analysis of the model output along iterations. The theoretical findings are supported by experiments.

Reviewer 2



This paper studies the implicit regularization of gradient descent over deep neural networks for deep matrix factorization models. The paper begins with a review of prior work regarding how running gradient descent on a shallow matrix factorization model, with small learning rate and initialization close to zero, tends to converge to solutions that minimize the nuclear norm [20] (Conjecture 1). This discussion is then extended to deep matrix factorization, where predictive performance improves with depth when the number of observed entries is small. Experimental results (Figure 2) which challenge Conjecture 1 are then presented, which indicate that implicit regularization in both shallow and deep matrix factorization converges to low-rank solutions, rather than minimizing nuclear norm, when few entries are observed. Finally, a theoretical and experimental analysis of the dynamics of gradient flow for deep matrix factorization is presented, which shows how singular values and singular vectors of the product matrix evolve during training, and how this leads to implicit regularization that induces low-rank solutions. The organization of the paper presents a narrative that evolves nicely as the paper progresses, which makes the paper relatively easy to follow. Most of the theoretical and experimental results are fairly convincing. However, some of the plots are a bit difficult to read, particularly in Figures 1 and 3, since some of the plots are missing axis labels. Also, it’s not clear how the plots in Fig. 3 can be interpreted to show that the singular values move slower when small and faster when large as depth is increased, as indicated by the authors. In Sec. 2.2, it is noted that when sufficiently many entries are observed, minimum nuclear norm and minimum rank are known to coincide. It would be helpful to provide further discussion or analysis regarding this point, perhaps by showing a plot of how the reconstruction error for the depth 2, 3, and 4 solutions compare to the minimum nuclear norm solution as a function of the number of observed entries.

Reviewer 3



Overall, I found this an extremely interesting paper. The paper is well written (though most proofs were in the Supplementary section), provides a good background (although this is also relegated to the Supplementary section), and makes an interesting contribution to our understanding of deep learning. I appreciated the approach as well, and at least to me, this was novel as well. The paper is original, technically sound (as far as I checked---but I cannot vouch for all the proofs), well written, and significant.

[Author Response · NeurIPS 2019]

We thank reviewers for their effort and support!

## Reviewer #1

We appreciate your feedback on Figures 1 and 2. Plots there were meant to emphasize trends as standard deviation of initialization tends to zero. We will consider deferring this information to supplementary material, and replacing the figures with tables that show results only for the minimal standard deviation.

## Reviewer #2

Thank you for your feedback on Figures 1 and 3. We will consider replacing Figure 1 with a table (see our response to Reviewer #1). As for Figure 3, we apologize for the rightmost plots not having titles and axis labels — this is a typo that will be corrected.

Text interpreting Figure 3 appears at the opening of Subsection 3.1. We expand on the plots for singular values (first three columns) herein. With depth 1 (first column) singular values show significant movement (as indicated by their slopes) right from the outset, until reaching stationarity. This results in a solution that does not have low rank (many large singular values). With depth 2 (second column) singular values initially (when close to zero) do not move much (flat slope), and then, when one gets far enough from zero, it begins to move fast (steep slope), until reaching stationarity. The resulting solution has low effective rank compared to depth 1 (a few large singular values and many smaller ones). Turning to depth 3 (third column), we see the same effect as with depth 2 but more potently — small singular values move slower (flatter slope) and those that escape the origin move faster (steeper slope). The final solution accordingly has lower effective rank — there are a few large singular values, and the rest are essentially at zero.

The fact that minimum nuclear norm and minimum rank coincide when sufficiently many entries are observed (and certain mild technical assumptions are met) was shown for example in Candès and Recht [2009]. We agree with you that a plot showing reconstruction errors as a function of the number of observed entries would be insightful. This will be added to the final version of the manuscript. Thank you for the suggestion!

## Reviewer #3

We deferred content to supplementary material primarily due to lack of space. We will consider replacing Figures 1 and 2 with tables (see our response to Reviewer #1), and using the available room for broadening our proof sketches. Thank you for your positive feedback!

## References

Emmanuel J Candès and Benjamin Recht. Exact matrix completion via convex optimization. *Foundations of Computational mathematics*, 9(6):717, 2009.


[Meta-Review · NeurIPS 2019]

The reviewers were unanimous in liking the paper. They make multiple presentational suggestions that the authors should incorporate in their final version: some figures are hard to read, as well as additional pointed discussion in certain sections such as Sec 2.2 (which the authors agreed to in their rebuttal).